# Drosulfakinin signaling modulates female sexual receptivity in *Drosophila*

Tao Wang[1,2,3†], Biyang Jing[4†], Bowen Deng[5], Kai Shi[2], Jing Li[6], Baoxu Ma[2], Fengming Wu[2]*, Chuan Zhou[2,3,6]*

[1]School of Life Sciences, University of Science and Technology of China, Hefei, China; [2]State Key Laboratory of Integrated Management of Pest Insects and Rodents Institute of Zoology, Chinese Academy of Sciences, Beijing, China; [3]University of Chinese Academy of Sciences, Beijing, China; [4]State Key Laboratory of Membrane Biology, College of Life Sciences, IDG/McGovern Institute for Brain Research, Peking-Tsinghua Center for Life Sciences, Academy for Advanced Interdisciplinary Studies, Center for Quantitative Biology, Academy for Advanced Interdisciplinary Studies, Peking University, Beijing, China; [5]Chinese Institute for Brain Research, Peking-Tsinghua Center for Life Sciences, Zhongguangcun Life Sciences Park, Beijing, China; [6]Institute of Molecular Physiology, Shenzhen Bay Laboratory, Shenzhen, China

**Abstract** Female sexual behavior as an innate behavior is of prominent biological importance for survival and reproduction. However, molecular and circuit mechanisms underlying female sexual behavior is not well understood. Here, we identify the Cholecystokinin-like peptide Drosulfakinin (DSK) to promote female sexual behavior in *Drosophila*. Loss of DSK function reduces female receptivity while overexpressing DSK enhances female receptivity. We identify two pairs of *Dsk*-expressing neurons in the central brain to promote female receptivity. We find that the DSK peptide acts through one of its receptors, CCKLR-17D3, to modulate female receptivity. Manipulation of CCKLR-17D3 and its expressing neurons alters female receptivity. We further reveal that the two pairs of *Dsk*-expressing neurons receive input signal from pC1 neurons that integrate sex-related cues and mating status. These results demonstrate how a neuropeptide pathway interacts with a central neural node in the female sex circuitry to modulate sexual receptivity.

*For correspondence: wufengming@ioz.ac.cn (FW); zhouchuan@ioz.ac.cn (CZ)

†These authors contributed equally to this work

Competing interest: The authors declare that no competing interests exist.

## Editor's evaluation

The manuscript by Wang and colleagues expands our understanding of the neural circuit mechanisms underpinning innate sexual behaviors in *Drosophila*. It exploits an arsenal of sophisticated tools to demonstrate that the neuropeptide Drosulfakinin (DSK) modulates female sexual receptivity via pC1-DSK-MP1-CCKLR-17D3 receptor expressing neurons. The study also introduces new transgenic tools that will be valuable for the community and will be of interest to neuroscientists exploring neuropeptide function and female sexual behavior.

## Introduction

Upon encountering a suitable courtship object, *Drosophila* males display a series of stereotypic courtship rituals, such as following the target, tapping, producing courtship song by extending a wing and vibrating it, licking, and attempting copulation (*Yamamoto and Koganezawa, 2013*). Yet, it is the female who decides whether to accept or reject the male based on her assessment of male courtship quality and her own readiness to mate (*Dickson, 2008*). Once the female is willing to accept a courting male, she would slow down and open her vaginal plate to allow copulation (*Ferveur, 2010*;

*Greenspan and Ferveur, 2000*; *Hall, 1994*). Conversely, the female rejects the male by extruding her ovipositor or flying away (*Cook and Connolly, 1973*; *Dickson, 2008*). Males and females play different roles in the sex life and take on different contribution in reproductive success. It is essential to understand and identify genetic and neural circuits that modulate innate sexual behavior. For male courtship, a number of genes controlling male courtship have been identified (*Billeter et al., 2002*; *Emmons and Lipton, 2003*) and corresponding neural circuits have been dissected (*Broughton et al., 2004*; *Clowney et al., 2015*; *Demir and Dickson, 2005*; *Kimura et al., 2008*; *Kohatsu et al., 2011*; *Pan and Baker, 2014*; *Ryner et al., 1996*; *Stockinger et al., 2005*; *Tanaka et al., 2017*; *Yamamoto and Koganezawa, 2013*; *Yu et al., 2010*), whereas molecular and circuit mechanisms underlying female sexual behavior are less clear.

In recent years, genetic studies have shown that several genes play critical roles in regulating female sexual behavior. For example, mutant females of *icebox* and *chaste* show lower mating success rates while mutant females of *pain* show higher mating success rates than wild-type females (*Carhan et al., 2005*; *Juni and Yamamoto, 2009*; *Kerr et al., 1997*; *Sakai et al., 2009*), and mutant females of *spinster* show enhanced rejection behavior (*Suzuki et al., 1997*). Moreover, specific subsets of neurons in the brain and ventral nerve cord are found to be involved in female sexual behavior. A significant decline of female sexual receptivity is observed when silencing specific neuron clusters in the central brain, such as two subsets of *doublesex*-expressing neurons (pCd and pC1) and two interneuron clusters (Spin-A and Spin-D) (*Sakurai et al., 2013*; *Zhou et al., 2014*). Female-specific vpoDNs in the brain integrate mating status and courtship song to control vaginal plate opening and female receptivity (*Wang et al., 2021*). Silencing either Abd-B neurons or SAG neurons located in the abdominal ganglion reduces female sexual receptivity (*Bussell et al., 2014*; *Feng et al., 2014*). In addition, female sexual behavior is also modulated by monoamines. In particular, dopamine not only plays a key role in regulating female sexual receptivity (*Neckameyer, 1998*), but also controls behavioral switching from rejection to acceptance in virgin females (*Ishimoto and Kamikouchi, 2020*); and octopamine is pivotal to female sexual behavior (*Rezával et al., 2014*). Neuropeptides including SIFamide and Mip are involved in female sexual receptivity (*Jang et al., 2017*; *Terhzaz et al., 2007*). Nevertheless, we know very little on how neuropeptides and peptidergic neurons control female sexual receptivity.

Drosulfakinin (DSK) is a neuropeptide, which is ortholog of Cholecystokinin (CCK) in mammals, and its two receptors (CCKLR-17D1 and CCKLR-17D3) have been identified in *Drosophila* (*Chen and Ganetzky, 2012*; *Kubiak et al., 2002*; *Nichols et al., 1988*; *Staljanssens et al., 2011*). Previous studies have revealed that DSK peptide is involved in multifarious regulatory functions including satiety/food ingestion (*Nässel and Williams, 2014*; *Söderberg et al., 2012*; *Williams et al., 2014b*), male courtship (*Wu et al., 2019*), and aggression (*Agrawal et al., 2020*; *Williams et al., 2014a*; *Wu et al., 2020*). However, whether DSK peptide and DSK neurons are crucial for female sexual behavior is not clear.

In this study, we find that DSK mutant females show reduced receptivity, and overexpression of DSK enhances female receptivity. We further show that DSK is crucial in two pairs of DSK neurons that function downstream of core sex-promoting neurons and upstream of CCKLR-17D3 neurons to modulate female sexual behavior. Our results reveal how the neuropeptide DSK functions in a subset of DSK neurons to interact with neural nodes in the sex circuity and acts through its receptor CCKLR-17D3 to control female sexual behavior.

## Results

### Neuropeptide DSK is crucial for virgin female receptivity

We previously found that neuropeptide DSK regulates intermale aggression in flies (*Wu et al., 2020*). To investigate the potential function of DSK in modulating female behaviors, we first monitored the change of virgin female receptivity in *Dsk* mutant (Δ*Dsk*), which was generated previously (*Wu et al., 2020*). In brief, the 5'-UTR and coding region were deleted by the CRISPR-Cas9 system (*Figure 1A*), which was validated by PCR (*Figure 1B and C*). No anti-DSK signal was detected in Δ*Dsk* female brains, whereas four pairs of neurons were detected in wide-type and Δ*Dsk/+* female brains by immunostaining with anti-DSK antibody (*Figure 1D*), which does not label the full set of DSK neurons as previously found (*Nichols and Lim, 1996*). Courtship chamber was used to examine mating behavior

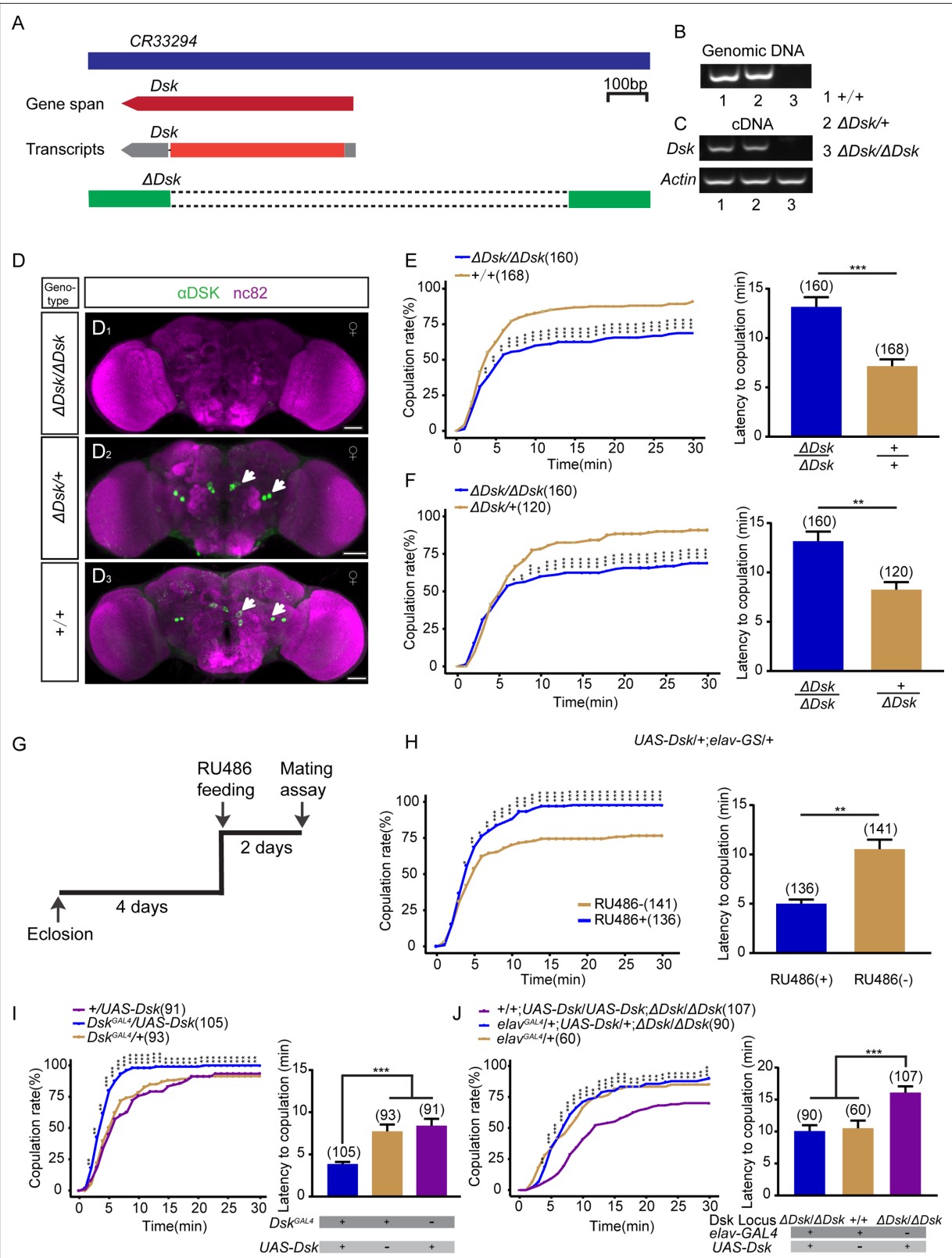

**Figure 1.** Drosulfakinin (*Dsk*) gene is important for female receptivity. (**A**) Organization of *Dsk* gene and generation of *ΔDsk*. (**B–C**) Validation of *ΔDsk*. PCR analysis at the deletion locus on genomic DNA samples of *ΔDsk/ΔDsk*, *+/ΔDsk*, *+/+*. (**B**) RT-PCR analysis from cDNA samples of *ΔDsk/ΔDsk*, *+/ΔDsk*, *+/+* (**C**). (**D**) Brain of indicated genotype, immunostained with anti-DSK antibody (green) and counterstained with nc82 (magenta). Arrows show cell bodies (green) stained with anti-DSK antibody. Scale bars, 50 µm. (**E–F**) Receptivity of virgin females within 30 min. *Dsk* mutant females reduced

*Figure 1 continued on next page*

*Figure 1 continued*

copulation rate and prolonged the latency to copulation compared with wild-type (**E**) and heterozygous females (**F**). (**G**) Schematic of experimental design. (**H**) Conditional overexpression of *Dsk* under the control of elav-GeneSwitch (*elav-GS*) significantly increased copulation rate and shortened the latency to copulation after feeding RU486 compared without feeding RU486. (**I**) Overexpression of *Dsk* in DSK neurons significantly increased copulation rate and shortened the latency to copulation compared with genetic controls. (**J**) Decreased female sexual behavior phenotypes of Δ*Dsk*/Δ*Dsk* were rescued by *elav*$^{GAL4}$ driving *UAS-Dsk*. The number of female flies paired with wild-type males is displayed in parentheses. For the copulation rate, chi-square test is applied. For the latency to copulation, Mann-Whitney U test is applied in (**E, F, and H**), Kruskal-Wallis and post hoc Mann-Whitney U tests are applied in (**I–J**). Error bars indicate SEM. *$p < 0.05$, **$p < 0.01$, ***$p < 0.001$.

The online version of this article includes the following source data and figure supplement(s) for figure 1:

**Source data 1.** Source data for *Figure 1B–C, D–E and H–J*.

**Figure supplement 1.** Behavior arena used in mating behavior assay.

**Figure supplement 2.** Mutation of Drosulfakinin (*Dsk*) in females did not alter female ovipositor extrusion.

**Figure supplement 2—source data 1.** Source data for *Figure 1—figure supplement 2B*.

**Figure supplement 3.** Effect of knocking down the expression of Drosulfakinin (*Dsk*) on female sexual behavior.

**Figure supplement 3—source data 1.** Source data for *Figure 1—figure supplement 3B-C,F-G*.

**Figure supplement 4.** Locomotor behavior of Drosulfakinin (*Dsk*) mutant and *Dsk* RNA interference (RNAi) in females.

**Figure supplement 4—source data 1.** Source data for *Figure 1—figure supplement 4*.

**Figure supplement 5.** Courtship behavior of wild-type males paired with females of indicated genotypes.

**Figure supplement 5—source data 1.** Source data for *Figure 1—figure supplement 5*.

**Figure supplement 6.** Drosulfakinin (*Dsk*) mutation or knockdown does not affect egg laying.

**Figure supplement 6—source data 1.** Source data for *Figure 1—figure supplement 6A, D*.

**Figure supplement 7.** Female receptivity of control flies for Drosulfakinin (Dsk) overexpression.

**Figure supplement 7—source data 1.** Source data for *Figure 1—figure supplement 7*.

(*Figure 1—figure supplement 1*), and two parameters including copulation rate and latency were used to characterize female receptivity (*Ferveur, 2010*). Interestingly, *Dsk* null mutant displayed reduced copulation rate and prolonged latency to copulation compared with wild-type (*Figure 1E*) and Δ*Dsk*/+ virgin females (*Figure 1F*). We asked whether the phenotype of decreased female receptivity in Δ*Dsk* flies is due to elevated rejection behaviors such as ovipositor extrusion, and found that Δ*Dsk* virgin females displayed similarly low levels of ovipositor extrusion like wild-type and Δ*Dsk*/+ virgin females (*Figure 1—figure supplement 2*).

To further confirm the decreased receptivity phenotype in Δ*Dsk* females, we knocked down the expression of *Dsk* using RNA interference (RNAi) under the control of a pan-neuronal *elav*$^{GAL4}$ driver, which significantly decreased DSK immunoreactivity (*Figure 1—figure supplement 3A, B*). We found that knocking down *Dsk* expression pan-neuronally significantly reduced female receptivity (*Figure 1—figure supplement 3C*). Furthermore, we also observed reduced female receptivity in females with *Dsk* knockdown using a knock-in *Dsk*$^{GAL4}$ generated previously (*Wu et al., 2020*; *Figure 1—figure supplement 3D-F*). This *Dsk*$^{GAL4}$ only labeled four pairs of neurons in the brain and no expression in the glia or gut (*Figure 1—figure supplement 3D-E*). It should be mentioned that this *Dsk*$^{GAL4}$ did not label insulin-producing cells (IPCs) in the PI region as previously found (*Nichols and Lim, 1996*). Thus, to investigate whether DSK peptide released from IPCs is involved in female sexual behavior, we knocked down the expression of *Dsk* only in these IPCs by using *Dilp2-GAL4* and found that restricting the expression of *DskRNAi* in IPCs did not affect virgin female receptivity (*Figure 1—figure supplement 3G*). No significant change of locomotor activity was detected in females with *Dsk* mutant or knockdown (*Figure 1—figure supplement 4*).

To investigate whether reduced copulation rate in Δ*Dsk* females is due to potential abatement of female sexual appeal, we examined courtship levels in wild-type males paired with Δ*Dsk* or control females and observed similarly high levels of courtship in all cases (*Figure 1—figure supplement 5*). Thus, the decreased receptivity in Δ*Dsk* females is not due to any change in male courtship efficiency, but rather a decline of willingness for copulation in these females.

As recently mated females may reduce receptivity and increase egg laying, we asked whether the decreased receptivity could be a post-mating response and correlate with elevated egg laying. To address this, we examined the number of eggs laid by virgin females with *Dsk* mutant or knockdown,

and found that manipulation of *Dsk* did not enhance egg laying in these virgin females (*Figure 1—figure supplement 6A*). To investigate whether DSK neurons respond to mating status, we measured the activity of these neurons using the transcriptional reporter of intracellular Ca$^{2+}$ (TRIC) in virgin and mated females. TRIC is designed to quantitatively monitor the change of neural activity by the reconstitution of a functional transcription factor in the presence of Ca$^{2+}$ (*Gao et al., 2015*). As mentioned above, four pairs of neurons were labeled by *Dsk$^{GAL4}$* driving the expression of *UAS-mCD8::GFP* (*Figure 1—figure supplement 3D*). However, we only observed TRIC signals in the two pairs of neurons in the middle area of female brains (*Figure 1—figure supplement 6B,C*). Quantification of these TRIC signals showed no significant difference in virgin and mated females (*Figure 1—figure supplement 6D*). These results further indicate that DSK neurons do not respond to mating status.

We next asked whether overexpression of *Dsk* would enhance virgin female receptivity. Conditional overexpression of *Dsk* under the control of elav-GeneSwitch (elav-GS), a RU486-dependent pan-neuronal driver (*Osterwalder et al., 2001*), induced copulation more quickly than control females without RU486 feeding (*Figure 1G and H*, *Figure 1—figure supplement 7*). In addition, overexpression of *Dsk* in DSK neurons using *Dsk$^{GAL4}$* also increased copulation rate and shortened latency to copulation compared with genetic control females (*Figure 1I*). Furthermore, we carried out genetic rescue experiments to further confirm the function of *Dsk* in modulating female sexual receptivity. To address this question, we used the pan-neuronal driver *elav$^{GAL4}$* to drive *UAS-Dsk* expression in *Dsk* mutant background, and found that neuron-specific expression of *Dsk* could restore the decreased receptivity in ΔDsk virgin females (*Figure 1J*). Taken together, these results indicate that the function of *Dsk* is crucial for female sexual receptivity, which also suggest that DSK neurons play a role in female sexual receptivity.

## DSK neurons promote virgin female receptivity

To further study how *Dsk*-expressing neurons regulate female sexual behavior, we first activated DSK neurons with *Dsk$^{GAL4}$* expressing the heat-activated *Drosophila* transient receptor potential channel (dTrpA1) (*Hamada et al., 2008*). Activation of DSK neurons increased virgin female receptivity at 29°C relative to 21°C (*Figure 2A*), whereas female receptivity was not changed between 29°C and 21°C in controls with either *UAS-dTrpA1* alone or *Dsk$^{GAL4}$* alone (*Figure 2B and C*). Meanwhile, we further analyzed whether activating DSK neurons would affect ovipositor extrusion in females with courting males and found that manipulation of DSK neurons did not affect ovipositor extrusion (*Figure 2—figure supplement 1*). We next tried to silence DSK neurons by using *Dsk$^{GAL4}$* to express tetanus toxin light chain (TNT), which blocks synaptic vesicle exocytosis (*Sweeney et al., 1995*), and found a significant reduction of receptivity in virgin females (*Figure 2D*). To test whether alteration of receptivity in females with DSK neurons activated or silenced is due to potential changes in general locomotion, we tested locomotor activity in individual females and found that the activating or silencing DSK neurons did not significantly affect locomotion (*Figure 2—figure supplement 2A,B*). Note that temperature shift did not affect female sexual behavior, although higher temperature induced higher locomotion velocity (*Soto-Padilla et al., 2018*). We further expressed an inwardly rectifier potassium channel (Kir2.1) that hyperpolarizes neurons and suppresses neural activity (*Baines et al., 2001*; *Thum et al., 2006*) in DSK neurons, and observed a decrease of virgin female receptivity (*Figure 2—figure supplement 3*).

Female receptivity depends on the female's sexual maturity and mating status. Very young virgins display low receptivity level to courting males and mated females become temporarily unreceptive to courting males (*Dickson, 2008*; *Rezával et al., 2012*). We tested whether activation of DSK neurons could also promote female sexual receptivity in very young virgins or mated females, and found that activation of DSK neurons did not alter female receptivity in either young virgins or mated females (*Supplementary file 1*). Together these results indicate that DSK neurons promote sexual behavior in virgin females.

## Two pairs of DSK-MP1 neurons promote virgin female receptivity

Analyses of the expression pattern of *Dsk$^{GAL4}$* revealed that four pairs of neurons were specifically labeled in the brain, which were classified into two types (two pairs of MP1 and two pairs of MP3) based on the location of cell bodies (*Nichols and Lim, 1996*; *Figure 1—figure supplement 3D*), and the two pairs of MP1 neurons were further classified into MP1a and MP1b based on the single-cell

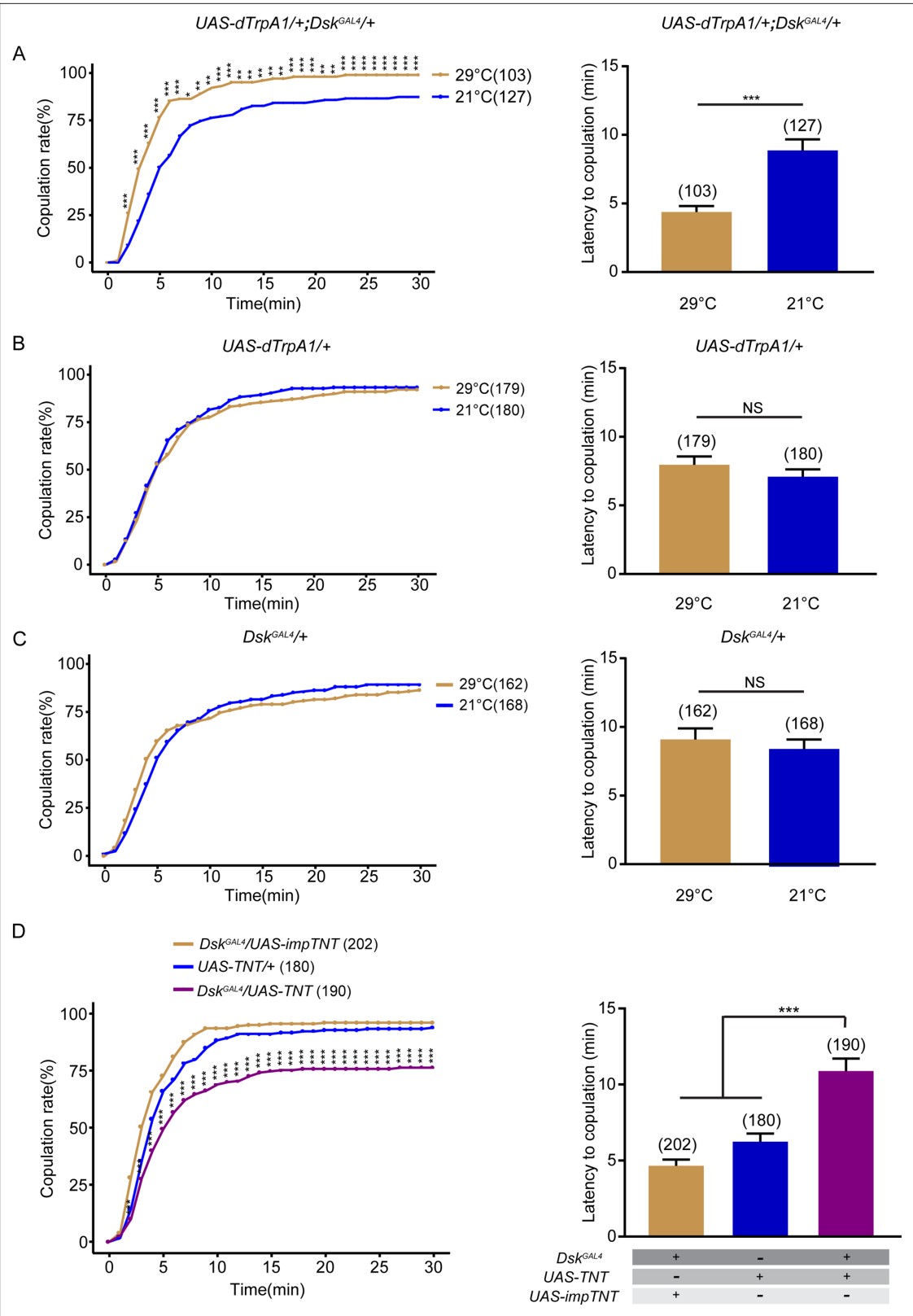

**Figure 2.** Drosulfakinin (DSK) neurons promote female receptivity. (**A**) Activation of DSK neurons significantly increased copulation rate and shortened the latency to copulation at 29°C relative to 21°C. *Dsk^GAL4* driving *UAS-dTrpA1* activated DSK neurons at 29°C. (**B–C**) The controls with either *UAS-dTrpA1* alone or *Dsk^GAL4* alone did not alter the copulation rate and the latency to copulation at 29°C relative to 21°C. (**D**) Inactivation of DSK neurons significantly decreased copulation rate and prolonged the latency to copulation compared with controls. *Dsk^GAL4* driving *UAS-TNT* inactivated DSK

*Figure 2 continued on next page*

*Figure 2 continued*

neurons. The number of female flies paired with wild-type males is displayed in parentheses. For the copulation rate, chi-square test is applied. For the latency to copulation, Mann-Whitney U test is applied in (**A–C**), Kruskal-Wallis and post hoc Mann-Whitney U tests are applied in (**D**). Error bars indicate SEM. *p < 0.05, **p < 0.01, ***p < 0.001, NS indicates no significant difference.

The online version of this article includes the following source data and figure supplement(s) for figure 2:

**Source data 1.** Source for *Figure 2*.

**Figure supplement 1.** Activating Drosulfakinin (DSK) neurons in females did not affect female ovipositor extrusion.

**Figure supplement 1—source data 1.** Source data for *Figure 2—figure supplement 1*.

**Figure supplement 2.** Locomotor behavior of females did not change after manipulating activity of Drosulfakinin (DSK) neurons.

**Figure supplement 2—source data 1.** Source data for *Figure 2—figure supplement 2*.

**Figure supplement 3.** Inactivation of Drosulfakinin (DSK) neurons reduced female receptivity.

**Figure supplement 3—source data 1.** Source data for *Figure 2—figure supplement 3*.

morphology of these neurons (*Wu et al., 2019*). However, the functional difference between MP1 and MP3 neurons was not characterized in female files due to the lack of genetic access.

To investigate whether one or both of the types are involved in regulating female sexual behavior, we used intersectional strategy to subdivide DSK neurons and manipulate DSK-MP1 and DSK-MP3 neurons separately. We screened ~100 knock-in GAL4 lines from the *Drosophila* chemoconnectome (CCT) project (*Deng et al., 2019*) combined with *DskFlp* to drive *UAS > stop > myr::GFP* (a Gal4/Flp-responsive membrane GFP reporter) expression, and further confirmed the identity of these neurons using the anti-DSK antibody. Interestingly, we found that intersection of *GluRIA$^{GAL4}$*, which targets glutamate receptor IA (GluRIA) cells, with *DskFlp* specifically labeled DSK-MP1 neurons (*Figure 3A*, *Figure 3—figure supplement 1A*), while intersection of *TβH$^{GAL4}$*, which targets octopaminergic neurons, with *DskFlp* specifically labeled DSK-MP3 neurons (*Figure 3B*, *Figure 3—figure supplement 1B*). Next, we investigated the behavioral relevance of specific subtypes of DSK neurons. Activation of DSK-MP1 neurons significantly increased virgin female receptivity (*Figure 3C*, *Figure 3—figure supplement 2A-C*), while inactivation of DSK-MP1 neurons significantly reduced virgin female receptivity (*Figure 3D*). In contrast, neither activation nor inactivation of DSK-MP3 neurons altered virgin female receptivity (*Figure 3E and F*, *Figure 3—figure supplement 2C-E*). Taken together, these results indicate that DSK-MP1 neurons, rather than DSK-MP3 neurons, play an essential role in regulating female sexual behavior.

## DSK regulates female receptivity via its receptor CCKLR-17D3

Next, we asked how DSK regulates female receptivity through its receptors. Two DSK receptors were previously identified: *CCKLR-17D1* and *CCKLR-17D3* (*Chen and Ganetzky, 2012*; *Kubiak et al., 2002*), and it would be essential to distinguish which receptor is or both of receptors are critical for modulating female sexual behavior. We first examined virgin female receptivity in either *CCKLR-17D1* mutant female, which was generated previously (*Wu et al., 2020*), or *CCKLR-17D1* RNAi knockdown female, and did not observe any effect on female receptivity (*Figure 4—figure supplement 1*). We then examined virgin female receptivity in *CCKLR-17D3* mutant female, which was also generated previously (*Wu et al., 2020*). In brief, the last four exons were deleted by the CRISPR-Cas9 system (*Figure 4A*), which was validated by PCR (*Figure 4B and C*). Interestingly, mutation of *CCKLR-17D3* reduced mating success rate in virgin females compared with wide-type and heterozygous control females (*Figure 4D*). Moreover, RNAi knockdown of *CCKLR-17D3* under the control of the pan-neuronal *elav$^{GAL4}$* driver or *CCKLR-17D3$^{GAL4}$* also significantly reduced female receptivity (*Figure 4—figure supplement 2A,B*). Conditional knockdown of *CCKLR-17D3* using the elav-GS system to avert the potential developmental effect also significantly decreased female receptivity (*Figure 4—figure supplement 2C-E*). In addition, no significant change of locomotor activity was detected in *CCKLR-17D3* mutant or knockdown females (*Figure 4—figure supplement 3*). Furthermore, the reduced female receptivity of *CCKLR-17D3* mutant females could be rescued by expression of *UAS-CCKLR-17D3* driven by *elav-GS* (*Figure 4E–G*). These results demonstrate that DSK acts through CCKLR-17D3 but not CCKLR-17D1 to promote female sexual receptivity.

To further determine whether CCKLR-17D3 neurons are functionally important for female sexual receptivity, we manipulated neurons labeled by the *CCKLR-17D3$^{GAL4}$*, which was generated previously

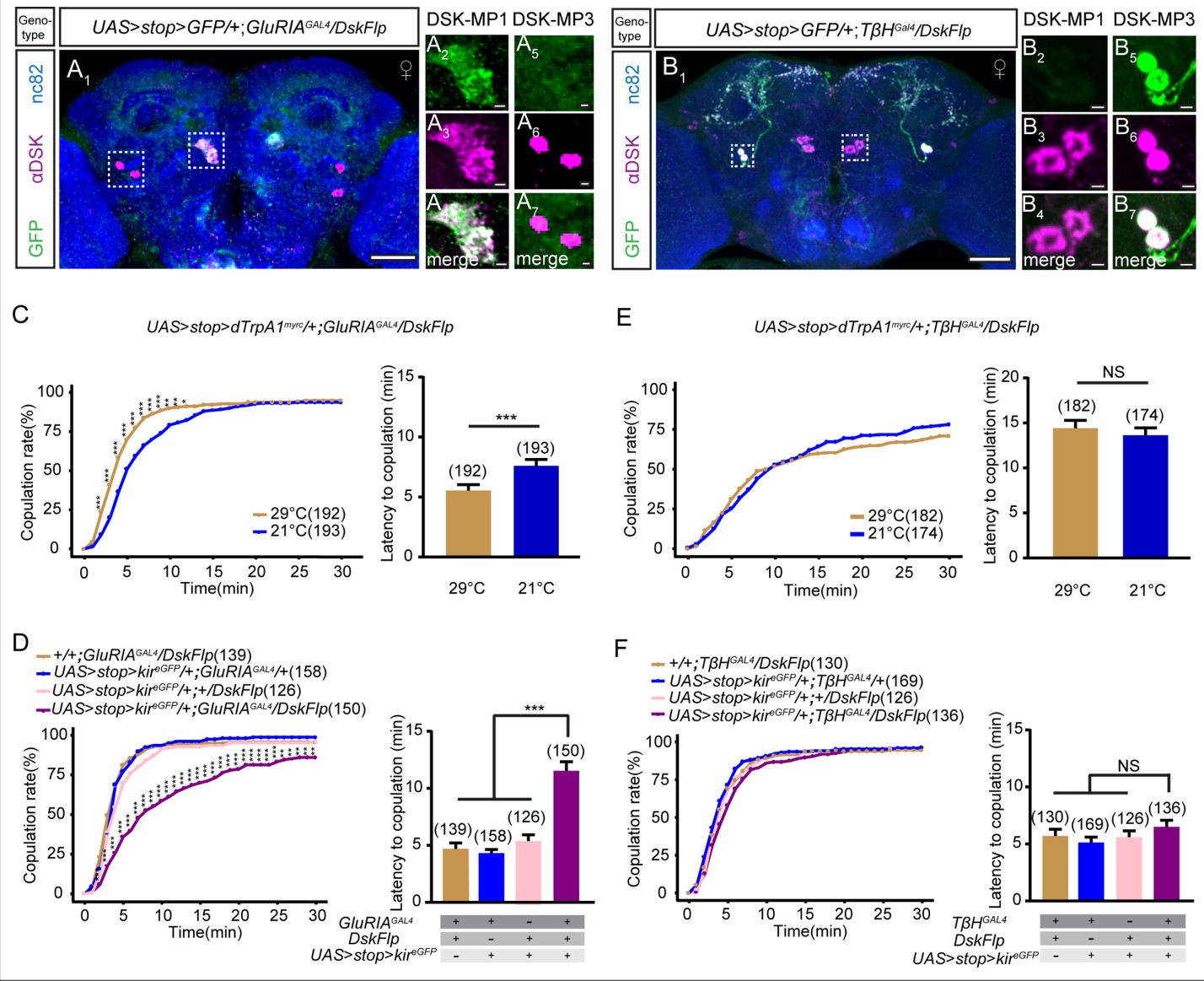

**Figure 3.** DSK-MP1 neurons play a critical role in female receptivity. (**A**) Intersectional expression of Drosulfakinin (*Dsk*) neurons and glutamate receptor IA (*GluRIA*) neurons were detected by immunostaining with anti-GFP (green) and anti-DSK (magenta) antibodies in female brain and were counterstained with anti-nc82 (blue). Magnification of white boxed region in (**A**) is shown in (**A₂–A₇**). Genotype: *UAS > stop > myr::GFP/+;GluRIA^GAL4/DskFlp*. (**B**) Intersectional expression of *Dsk* neurons and *TβH* neurons were detected by immunostaining with anti-GFP (green) and anti-DSK (magenta) antibodies in female brain and were counterstained with anti-nc82 (blue). Magnification of white boxed region in (**B**) is shown in (**B₂–B₇**). Genotype: *UAS > stop > myr::GFP/+;TβH^GAL4/DskFlp*. Scale bars are 50 µm in (**A₁ and B₁**), 5 µm in (**A₂–A₇**) and (**B₂–B₇**). (**C**) Activation of co-expression neurons of *Dsk* and *GluRIA* significantly increased copulation rate and shortened the latency to copulation at 29°C relative to 21°C. Genotype: *UAS > stop > dTrpA^myrc/+;GluRIA^GAL4/DskFlp*. (**D**) Inactivation of co-expression neurons of *Dsk* and *GluRIA* significantly decreased the copulation rate and prolonged the latency to copulation compared with controls. Genotype: *UAS > stop > kir^eGFP/+;GluRIA^GAL4/DskFlp, +/+;GluRIA^GAL4/DskFlp, UAS > stop > kir^eGFP/+;GluRIA^GAL4/+, UAS > stop > kir^eGFP/+;+/DskFlp*. (**E**) Activation of co-expression neurons of *Dsk* and *TβH* did not alter the copulation rate and copulation latency at 29°C relative to 21°C. Genotype: *UAS > stop > dTrpA^myrc/+;TβH^GAL4/DskFlp*. (**F**) Inactivation of co-expression neurons of *Dsk* and *TβH* did not alter the copulation rate and copulation latency compared with controls. Genotype: *UAS > stop > kir^eGFP/+;TβH^GAL4/DskFlp, UAS > stop > kir^eGFP/+;+/DskFlp, UAS > stop > kir^eGFP/+;TβH^GAL4/+, +/+;TβH^GAL4/DskFlp*. The number of female flies paired with wild-type males is displayed in parentheses. For the copulation rate, chi-square test is applied. For the latency to copulation, Mann-Whitney U test is applied in (**C and E**), Kruskal-Wallis and post hoc Mann-Whitney U tests are applied in (**D and F**). Error bars indicate SEM. *p < 0.05, **p < 0.01, ***p < 0.001, NS indicates no significant difference.

The online version of this article includes the following source data and figure supplement(s) for figure 3:

**Source data 1.** Source for *Figure 3*.

*Figure 3 continued on next page*

*Figure 3 continued*

**Figure supplement 1.** Expression pattern of Drosulfakinin (DSK) neurons targeted with genetic intersections in the ventral nerve cord.

**Figure supplement 2.** Female receptivity of control females for activation of specific Drosulfakinin (DSK) neurons.

**Figure supplement 2—source data 1.** Source for *Figure 3—figure supplement 2*.

(*Wu et al., 2020*). The *CCKLR-17D3^GAL4* labeled neuronal clusters in the central complex, SOG, and ventral nerve cord (*Figure 4—figure supplement 4A*). We activated *CCKLR-17D3^GAL4* neurons using *dTrpA1* and observed significantly increased mating success rate in virgin females at 29°C than 21°C (*Figure 4H*), whereas female receptivity was not changed between 29°C and 21°C in control females (*Figure 4I*). Moreover, we also inactivated *CCKLR-17D3^GAL4* neurons by expressing TNT and found that female receptivity was decreased after inactivating these neurons (*Figure 4J*). Thus, *CCKLR-17D3^GAL4* neurons positively regulate virgin female receptivity.

It has been well established that *doublesex (dsx)* expressing neurons play a key role in regulating female sexual behavior (*Feng et al., 2014*; *Rideout et al., 2010*; *Wang et al., 2021*; *Zhou et al., 2014*). Thus, we asked whether *CCKLR-17D3^GAL4* drives expression in *dsx* neurons to regulate female receptivity. However, intersection between *CCKLR-17D3^GAL4* and *dsx^LexA* only labeled projections from peripheral sensory neurons that innervate the SOG region (*Figure 4—figure supplement 4B*). Furthermore, either overexpressing or knocking down CCKLR-17D3 in all *dsx* neurons did not alter virgin female receptivity (*Figure 4—figure supplement 4C,D*). These results indicate that CCKLR-17D3 did not function in *dsx* neurons to regulate female sexual behavior.

To further confirm whether CCKLR-17D3 is the downstream target of DSK on female receptivity, we tested receptivity in females with DSK neurons activated by dTrpA1 under the *CCKLR-17D3* mutant background. We found that loss of CCKLR-17D3 function could block the increased levels of female receptivity caused by activating DSK neurons (*Figure 4K–M*). Together these results demonstrate that DSK released from DSK-MP1 neurons acts on its receptor CCKLR-17D3 to promote female sexual receptivity.

## DSK neurons function downstream of the sex-promoting *R71G01-GAL4* neurons

In males, *R71G01-GAL4* drives the expression of P1 neurons that interact with DSK neurons to regulate male courtship (*Wu et al., 2019*) and aggression (*Wu et al., 2020*). Previous studies employed the intersection of *R71G01-LexA* with *dsx^GAL4* to specifically label and manipulate pC1 neurons, which integrate male courtship and pheromone cues to promote virgin female receptivity (*Wang et al., 2020*; *Zhou et al., 2014*). We found that activation of *R71G01-GAL4* neurons consisting of pC1 and a few other neurons promoted female receptivity (*Figure 5—figure supplement 1*), similarly as previously activating pC1 neurons using the intersectional strategy (*Zhou et al., 2014*). Thus, we asked whether DSK neurons would interact with *R71G01-GAL4* neurons to control female sexual behavior. To address this question, we first sought to detect whether *Dsk*-expressing neurons had potential synaptic connection with *R71G01-GAL4* neurons via GFP reconstitution across synaptic partners (GRASP) (*Feinberg et al., 2008*; *Gordon and Scott, 2009*). Interestingly, we detected significant reconstituted GFP signals between *R71G01-LexA* and *Dsk^GAL4* labeled neurons (*Figure 5—figure supplement 2*), suggesting that these neurons might have synaptic connection. Next, we surveyed whether *Dsk*-expressing neurons are immediate downstream of *R71G01-GAL4* neurons by using *trans*-Tango, a method of anterograde transsynaptic tracing (*Talay et al., 2017*). Interestingly, *R71G01-GAL4* downstream *trans*-Tango signals were observed in DSK neurons by co-staining the *trans*-Tango flies with the anti-DSK antibody (*Figure 5A and B*, *Figure 5—figure supplement 3*). Moreover, we registered *R71G01-GAL4* neurons and DSK neurons, and found that axons of *R71G01-GAL4* neurons partly overlapped with dendrites of DSK neurons (*Figure 5C*).

In addition, we performed behavioral epistasis experiment to confirm functional interactions between DSK neurons and *R71G01-GAL4* neurons. We activated *R71G01-GAL4* neurons by dTrpA1 in the *Dsk* mutant background, and found that increased levels of female receptivity caused by activation of *R71G01-GAL4* neurons were suppressed by the mutation in *Dsk* (*Figure 5D–G*). Taken together, these results further demonstrate that DSK neurons are the functional targets of *R71G01-GAL4* neurons in controlling female sexual behavior.

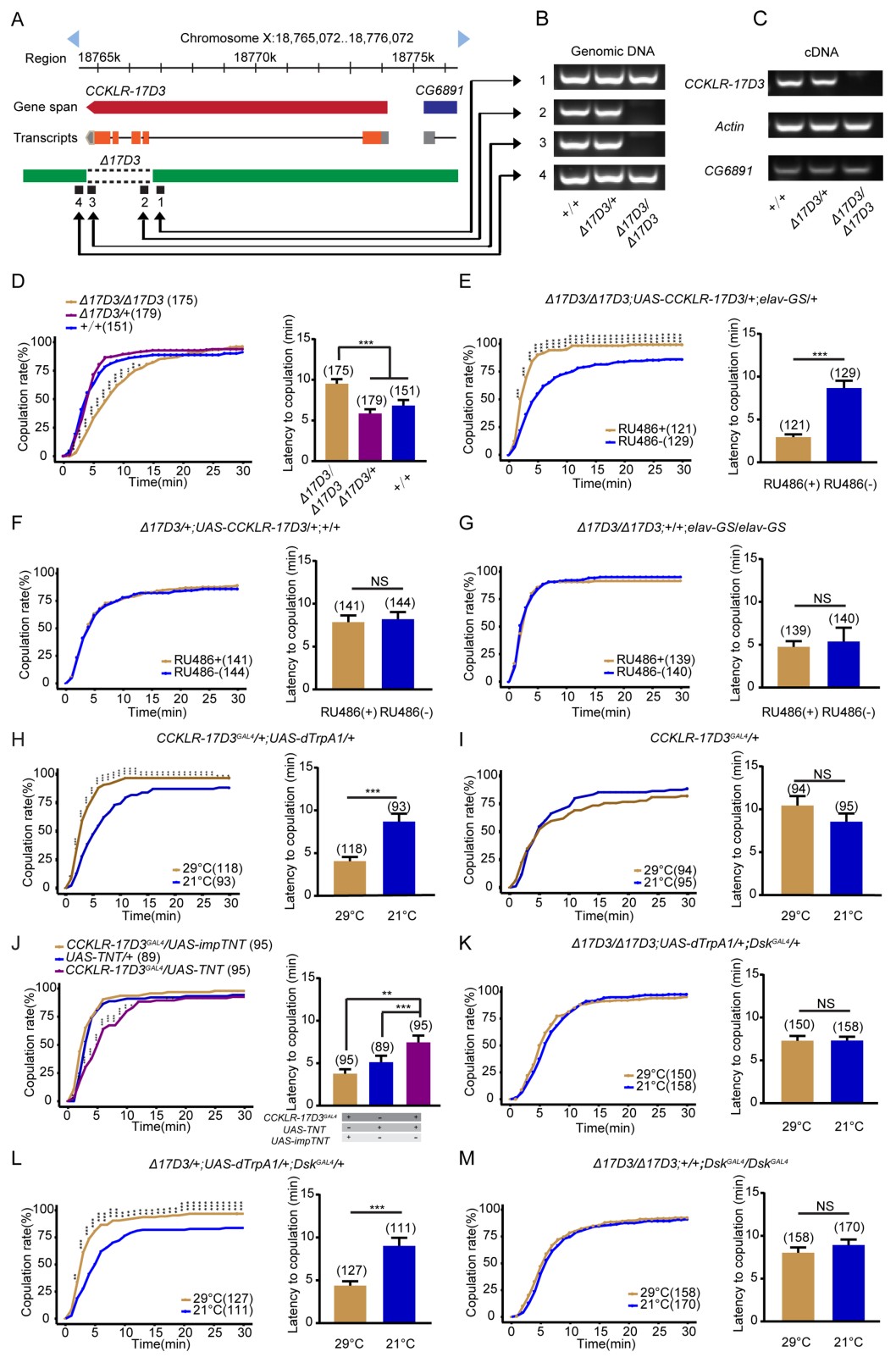

**Figure 4.** Drosulfakinin (*Dsk*) regulates female receptivity via *CCKLR-17D3* receptor. (**A**) Organization of *CCKLR-17D3* and generation of *Δ17D3*. (**B–C**) Validation of *Δ17D3*. PCR analysis from genomic DNA samples of *Δ17D3/Δ17D3*, +/*Δ17D3*, +/+ (**B**). RT-PCR analysis from cDNA samples of *Δ17D3/Δ17D3*, +/*Δ17D3*, +/+ (**C**). (**D**) *CCKLR-17D3* mutant females significantly decreased copulation rate and prolonged the latency to copulation compared

*Figure 4 continued on next page*

*Figure 4 continued*

with wild-type and heterozygous. (**E**) Conditional expression of *UAS-CCKLR-17D3* in the *Δ17D3* mutant background after feeding RU486 significantly increased copulation rate and shortened the latency to copulation compared without feeding RU486. (**F–G**) The controls with either *UAS-CCKLR-17D3* alone or elav-GeneSwitch (*elav-GS*) alone did not rescue the phenotypes of *Δ17D3/Δ17D3* at feeding RU486 relative to without feeding RU486. (**H**) Activating CCKLR-17D3 neurons significantly increased copulation rate and shortened the latency to copulation at 29°C relative to 21°C. *CCKLR-17D3^GAL4* driving *UAS-dTrpA1* activated CCKLR-17D3 neurons at 29°C. (**I**) The control with *CCKLR-17D3^GAL4* alone did not alter the copulation rate and the latency to copulation at 29°C relative to 21°C. (**J**) Inactivation of CCKLR-17D3 neurons significantly decreased copulation rate and prolonged the latency to copulation compared with controls. *Dsk^GAL4* driving *UAS-TNT* inactivated DSK neurons. (**K**) The copulation rate and the latency to copulation have no difference at 29°C relative to 21°C in the case of activating DSK neurons in the *Δ17D3* mutant background. (**L**) The positive control significantly increased copulation rate and shortened the latency to copulation at 29°C relative to 21°C. (**M**) The negative control did not alter the copulation rate and the latency to copulation by heating. The number of female flies paired with wild-type males is displayed in parentheses. For the copulation rate, chi-square test is applied. For the latency to copulation, Kruskal-Wallis and post hoc Mann-Whitney U tests are applied in (**D and J**), Mann-Whitney U test is applied in (**E–I and K–M**). Error bars indicate SEM. *$p < 0.05$, **$p < 0.01$, ***$p < 0.001$, NS indicates no significant difference.

The online version of this article includes the following source data and figure supplement(s) for figure 4:

**Source data 1.** Source data for *Figure 4*.

**Figure supplement 1.** *CCKLR-17D1* mutant or knockdown does not affect female receptivity.

**Figure supplement 1—source data 1.** Source data for *Figure 4—figure supplement 1*.

**Figure supplement 2.** *CCKLR-17D3* knockdown reduces female receptivity.

**Figure supplement 2—source data 1.** Source data for *Figure 4—figure supplement 2*.

**Figure supplement 3.** Locomotor behavior of *CCKLR-17D3* mutant and knockdown females.

**Figure supplement 3—source data 1.** Source data for *Figure 4—figure supplement 3*.

**Figure supplement 4.** CCKLR-17D3 neurons do not overlap with *doublesex (dsx)* neurons.

**Figure supplement 4—source data 1.** Source data for *Figure 4—figure supplement 4C-D*.

As the *R71G01-GAL4* labels pC1 neurons as well as a few other neurons, we further utilized the recently generated pC1 splitGAL4 drivers (*Wang et al., 2020*). We registered pC1 neurons labeled by two independent pC1 splitGAL4 (*pC1-ss1* and *pC1-ss2*) with DSK neurons, and found that axons of pC1 neurons overlapped with dendrites of DSK neurons (*Figure 5—figure supplement 4*). Furthermore, we utilized the recently generated full adult female brain (FAFB) electron microscopic (EM) image set (*Scheffer et al., 2020*), and found that pC1 neurons have intense synaptic input on DSK-MP1 neurons, especially the single pair of DSK-MP1b neurons, and few input on DSK-MP3 neurons (*Supplementary file 2*). These results indicate that DSK neurons are direct targets of *R71G01-GAL4* labeled pC1 neurons.

## Functional connectivity between pC1 neurons and DSK neurons

The above results showed that DSK neurons act downstream of *R71G01-GAL4* labeled pC1 neurons to promote female sexual receptivity. To further reveal the functional connectivity between pC1 neurons and *Dsk*-expressing neurons, we activated all *R71G01-GAL4* neurons through ATP activation of ATP-gated $P2X_2$ channel (*Brake et al., 1994*; *Yao et al., 2012*) and recorded the electrical responses in DSK-MP1 neurons and DSK-MP3 neurons using patch clamp (*Figure 6A*). In perforate patch recordings, ATP/$P2X_2$ activation of *R71G01-GAL4* neurons induced strong electrical responses from DSK-MP1 neurons and relatively weaker responses from DSK-MP3 neurons in female brains (*Figure 6B and C*). Thus, these results together with the above EM data unambiguously demonstrate that *Dsk*-expressing DSK-MP1 neurons receive input from sex-promoting pC1 neurons.

## Discussion

In this study, we systematically investigated DSK-mediated neuromodulation of female sexual receptivity. At the molecular level, we revealed that DSK neuropeptide and its receptor CCKLR-17D3 are crucial for modulating female sexual receptivity. At the neuronal circuit level, we identified that DSK

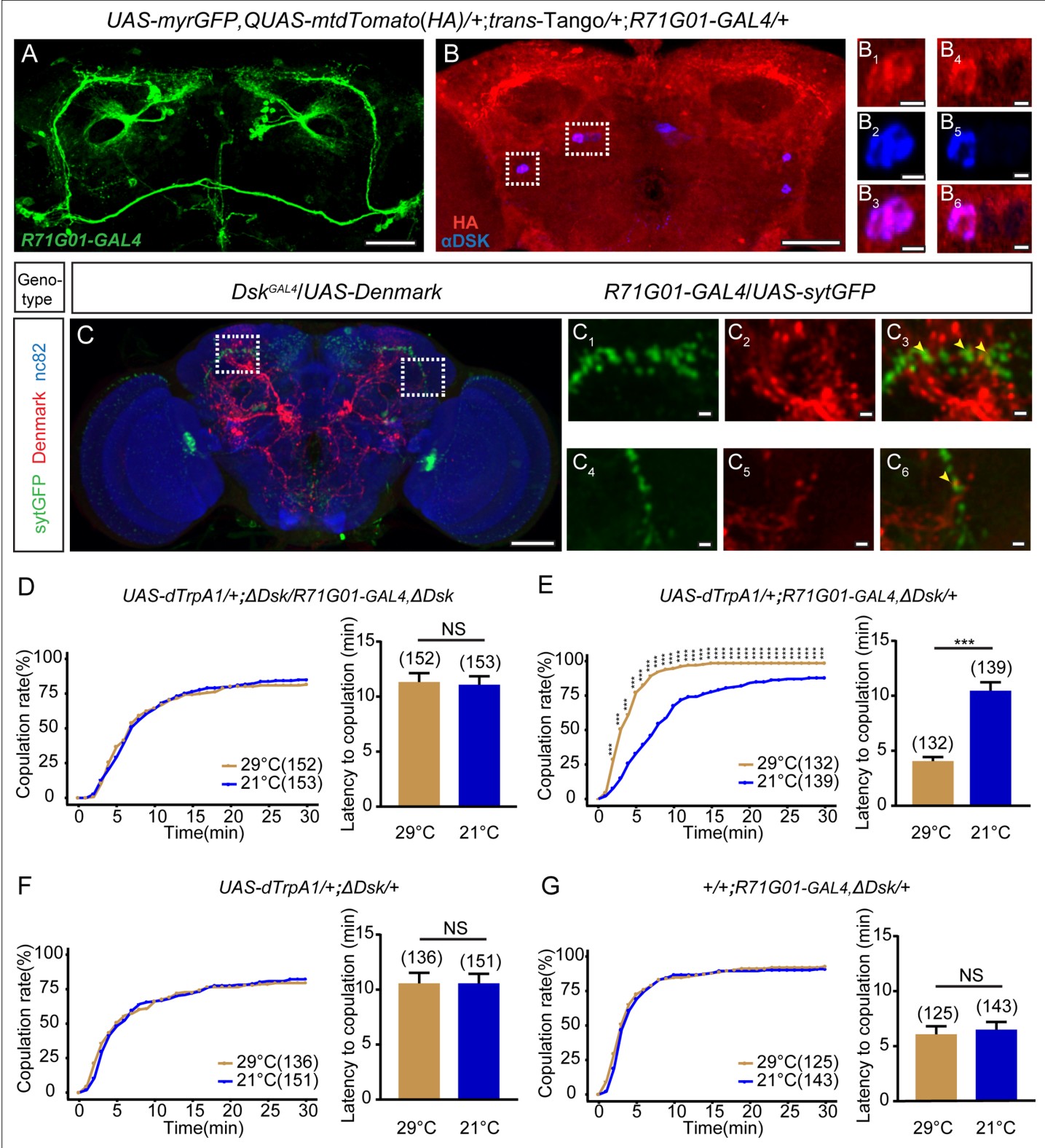

**Figure 5.** Drosulfakinin (DSK) neurons are functional targets of *R71G01-GAL4* neurons in regulating mating behavior. (**A–B**) Transsynaptic circuit analysis using *trans*-Tango confirms that *Dsk*-expressing neurons are postsynaptic neurons of *R71G01-GAL4* neurons. In the central brain, expression of the Tango ligand in *R71G01-GAL4* neurons (green) (**A**) induced postsynaptic mtdTomato signals (anti-HA, red) (**B**). Cell bodies of Dsk were stained with anti-DSK (blue) (**B**). Magnification of white boxed region in (**B**) is shown in (**B₁–B₃**) and (**B₄–B₆**). Scale bars are 50 μm in (**A–B**), 5 μm in (**B₁–B₃**) and (**B₄–B₆**). (**C**) Axons of *R71G01-GAL4* neurons overlapped with dendrites of DSK neurons by anatomical registration. Magnification of white boxed region

*Figure 5 continued on next page*

*Figure 5 continued*

in (**C**) is shown in (**C₁**–**C₃**) and (**C₄**–**C₆**). Yellow arrowheads indicated the region of overlaps between *R71G01-GAL4* neurons axons with DSK neurons dendrites. *R71G01-GAL4*-driven *UAS-sytGFP* expression (green), *Dsk^GAL4*-driven *UAS-Denmark* expression (red). Scale bars are 50 μm in (**C**), 5 μm in (**C₁**–**C₃**) and (**C₄**–**C₆**). (**D**) The copulation rate and the latency to copulation had no difference at 29°C relative to 21°C in the case of activation of *R71G01-GAL4* neurons in the *ΔDsk* mutant background. (**E**) The positive control significantly increased copulation rate and shortened the latency to copulation at 29°C relative to 21°C. (**F**–**G**) The negative controls did not alter the copulation rate and the latency to copulation by heating. The number of female flies paired with wild-type males is displayed in parentheses. For the copulation rate, chi-square test is applied. For the latency to copulation, Mann-Whitney U test is applied. Error bars indicate SEM. ***p < 0.001, NS indicates no significant difference.

The online version of this article includes the following source data and figure supplement(s) for figure 5:

**Source data 1.** Source data for *Figure 5*.

**Figure supplement 1.** Activation of *R71G01-GAL4* neurons promotes female receptivity.

**Figure supplement 1—source data 1.** Source data for *Figure 5—figure supplement 1*.

**Figure supplement 2.** Potential connection between *R71G01* neurons and Drosulfakinin (DSK) neurons was detected by GFP reconstitution across synaptic partners (GRASP) method.

**Figure supplement 3.** Control signals for the *trans*-Tango experiment.

**Figure supplement 4.** Axons of pC1 neurons overlapped with dendrites of Drosulfakinin (DSK) neurons.

neurons are the immediate downstream targets of sex-promoting pC1 neurons in controlling female sexual receptivity. Moreover, we employed intersectional tools to subdivide DSK neurons into medial DSK neurons (DSK-MP1) and lateral DSK neurons (DSK-MP3) and uncovered that DSK-MP1 neurons rather than DSK-MP3 neurons play essential roles in modulating female receptivity. Collectively, our findings illuminate a pC1-DSK-MP1-CCKLR-17D3 pathway that modulates female sexual behaviors in *Drosophila*.

The female sexual behavior is a complex innate behavior. The decision for the female to accept a courting male or not depends on not only sensory stimulation but also internal states. If the female is willing to mate, she slows down, pauses, and opens her vaginal plates to accept a courting male (**Bussell et al., 2014**; **Laturney and Billeter, 2014**; **Wang et al., 2021**), if not, she extrudes her ovipositor to deter a courting male or flies away (**Cook and Connolly, 1973**). Our results show that DSK signaling is crucial for virgin female receptivity but has no effect on ovipositor extrusion behavior. How exactly does the DSK signaling regulate virgin female receptivity is still not clear. One possibility is that DSK signaling regulates pausing behavior in response to male courtship (**Bussell et al., 2014**), as DSK receptor CCKLR-17D3 expresses in the central complex that has been found to be crucial for locomotor behaviors (**Strauss, 2002**). However, we did not observe any change in locomotor activity in DSK or CCKLR-17D3 mutant females. Note that we only assayed locomotor behavior in single females but not in females paired with courting males due to technical limit for analysis, and it is possible that the DSK signaling does not affect general locomotor behavior but regulates courtship-stimulated pausing behavior.

The four pairs of DSK neurons are classified into two types (DSK-MP1 and DSK-MP3) based on the location of the cell bodies, and DSK-MP1 neurons extend descending fibers to ventral nerve cord (**Wu et al., 2020**). In this study, we also found that activating DSK-MP1 neurons enhance female receptivity whereas inactivating DSK-MP1 neurons reduce female receptivity. Silencing adult Abd-B neurons and SAG neurons located in the abdominal ganglion inhibits female sexual receptivity (**Bussell et al., 2014**; **Feng et al., 2014**). It has been found that Abd-B neurons control female pausing behavior, and it would be interesting to further investigate whether DSK-MP1 neurons relay information from Abd-B neurons to regulate pausing and receptivity in females. We also note that DSK-MP1 neurons extend projections to suboesophageal ganglion (SOG), and the SOG is the terminus of ascending projections from a subset of female reproductive tract sensory neurons labeled by *pickpocket (ppk)*, *fruitless (fru)*, and *doublesex (dsx)* (**Häsemeyer et al., 2009**; **Rezával et al., 2012**; **Yang et al., 2009**). It is also possible that DSK-MP1 neurons may integrate information directly from these sensory neurons to regulate female receptivity. Another study further classified the DSK-MP1 neurons into two types (MP1a and MP1b) based on the morphology of their neuritis (**Wu et al., 2019**). Future studies would further build genetic tools to uncover the function of each subset of DSK neurons in regulating distinct innate behaviors, such as male courtship (**Wu et al., 2019**), aggression (**Agrawal et al., 2020**; **Wu et al., 2020**), and female sexual behavior.

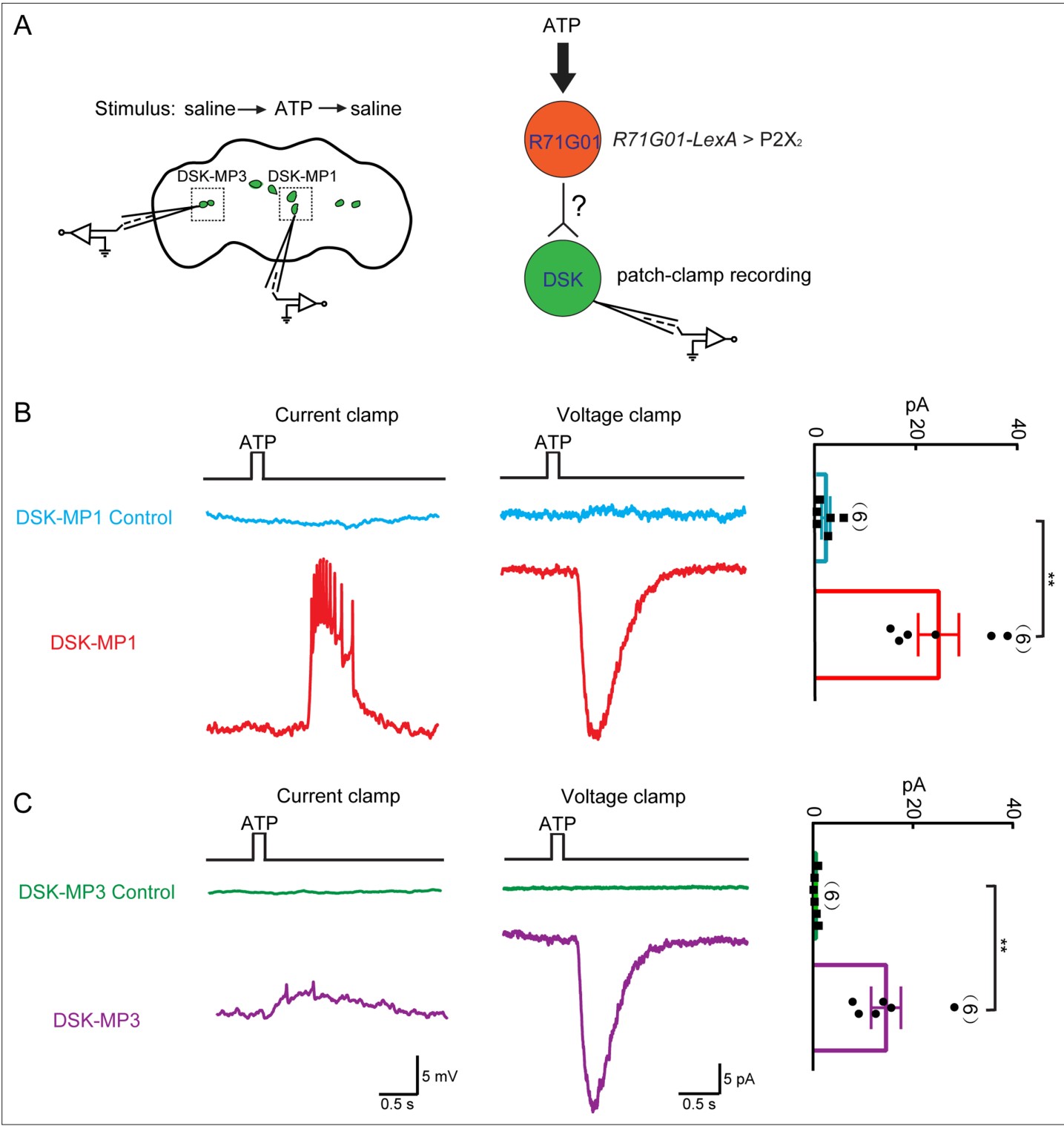

**Figure 6.** Functional connectivity between *R71G01-GAL4* neurons and Drosulfakinin (DSK) neurons. (**A**) Left: ATP stimulation and recording arrangement. The chemical stimulation is implemented using a three-barrel tube (with the tip positioned ~50 µm away from the brain), controlled by a stepper for rapid solution change. Right: schematic illustrating the activation of *R71G01-GAL4* neurons by ATP and patch-camp recording of DSK neurons. *R71G01-GAL4* neurons were activated by ATP in *+/+;R71G01-LexA/+;Dsk^GAL4/LexAop-P2X_2,UAS-GCaMP6m* files. (**B–C**) The electrical responses of medial DSK neurons (DSK-MP1) and lateral DSK neurons (DSK-MP3) to the ATP activation of 2X_2-expressing *R71G01-GAL4* neurons. ATP: 2.5 mM. Left: ATP-induced spiking firing (current clamp). Middle: current responses (voltage clamp). Right: quantification of absolute current responses. n = 6 for

*Figure 6 continued on next page*

*Figure 6 continued*

DSK-MP1, DSK-MP1 control, DSK-MP3, DSK-MP3 control. Genotype: *+/+;+/+;Dsk^GAL4/LexAop-P2X₂,UAS-GCaMP6m* for DSK-MP1 control and DSK-MP3 control. **p < 0.01 (Mann-Whitney U tests).

The online version of this article includes the following source data for figure 6:

**Source data 1.** Source for *Figure 6*.

Previous studies have revealed that pC1 neurons extend projections to lateral protocerebral complex (LPC) and this neural cluster responds to courtship song and cVA (*Zhou et al., 2015*; *Zhou et al., 2014*). Moreover, recent works have shown that DSK-MP1 neurons project to the region of LPC (*Wu et al., 2020*; *Wu et al., 2019*). We used GRASP, trans-Tango, and patch-clamp techniques and revealed that DSK-MP1 neurons are direct downstream target of *R71G01-GAL4* neurons that include pC1. EM reconstruction revealed that pC1 neurons have intense synaptic input on MP1b but not MP1a neurons, suggesting a crucial role of the single pair of MP1b neurons in female receptivity. Based on these findings, we propose that: (1) pC1 neurons act as a central node for female sexual receptivity by integrating sex-related sensory cues (courtship song and cVA) and mating status; and (2) DSK-MP1 neurons may integrate internal states (*Wu et al., 2019*) and pC1-encoded information to modulate female sexual behavior. Thus, it is of prime importance to further investigate how a neuropeptide pathway modulate a core neural node in the sex circuitry to fine-tune the female's willingness for sexual behavior in the future.

# Materials and methods

## Key resources table

| Reagent type (species) or resource | Designation | Source or reference | Identifiers | Additional information |
|---|---|---|---|---|
| Antibody | Mouse anti- a-bruchpilot monoclonal (nc82) | Developmental Studies Hybridoma Bank | Cat# nc82, RRID:AB_2314866 | IHC (1:50) |
| Antibody | Rat monoclonal anti-HA | Roche | Cat# 11867431001, RRID:AB_390919 | IHC (1:100) |
| Antibody | Mouse monoclonal anti-GFP-20 | Sigma-Aldrich | Cat# G6539, RRID:AB_259941 | IHC (1:100) |
| Antibody | Chicken polyclonal anti-GFP | Thermo Fisher Scientific | Cat# A10262, RRID:AB_2534023 | IHC (1:1000) |
| Antibody | Goat anti-chicken polyclonal, Alexa Fluor 488 | Thermo Fisher Scientific | Cat# A-11039; RRID: AB_2534096 | IHC (1:500) |
| Antibody | Goat anti-rat polyclonal, Alexa Fluor 546 | Thermo Fisher Scientific | Cat# A-11081, RRID:AB_2534125 | IHC (1:500) |
| Antibody | Goat anti-rabbit polyclonal, Alexa Fluor 546 | Thermo Fisher Scientific | Cat# A-11010, RRID:AB_2534077 | IHC (1:500) |
| Antibody | Goat anti-mouse polyclonal, Alexa Fluor 633 | Thermo Fisher Scientific | Cat# A-21094, RRID:AB_2535749 | IHC (1:500) |
| Antibody | Goat anti-rabbit polyclonal, Alexa Fluor 647 | Thermo Fisher Scientific | Cat# A-21247, RRID:AB_141778 | IHC (1:500) |
| Antibody | Rabbit polyclonal anti-DSK | | N/A | IHC(1:1000) |
| Chemical compound, drug | Paraformaldehyde (PFA) | Electron Microscopy Sciences | Cat# 15713 | 8% PFA diluted in 1× PBS at 1:4 or 1:2 |
| Chemical compound, drug | DPX Mountant | Sigma-Aldrich | Cat# 44581 | |
| Chemical compound, drug | Normal goat serum | Sigma-Aldrich | Cat# G9023 | |
| Chemical compound, drug | Adenosine 5'-triphosphate disodium salt hydrate microbial | Sigma-Aldrich | Cat# A6419-1G | 2.5 mM |

*Continued on next page*

*Continued*

| Reagent type (species) or resource | Designation | Source or reference | Identifiers | Additional information |
|---|---|---|---|---|
| Chemical compound, drug | Mifepristone (RU486) | Sigma-Aldrich | Cat# M8046-1G | |
| Genetic reagent (*Drosophila melanogaster*) | UAS-myrGFP,QUAS-mtdTomato(3*HA);trans-Tango | Zhong Lab, Tsinghua University | N/A | |
| Genetic reagent (*Drosophila melanogaster*) | +; sp/cyo; LexAop-P2X2, UAS-GCamP/Tm2 | Luo Lab, Peking University | N/A | |
| Genetic reagent (*Drosophila melanogaster*) | UAS-mCD8::GFP | Bloomington Stock Center | # 5137 | |
| Genetic reagent (*Drosophila melanogaster*) | 10XUAS-IVS-mCD8::RFP,13XLexAop2-mCD8::GFP; nSyb-MKII::nlsLexADBD/CyO; UAS-p65AD::CaM | Bloomington Stock Center | # 61679 | |
| Genetic reagent (*Drosophila melanogaster*) | TβH-GAL4 | Bloomington Stock Center | # 45904 | |
| Genetic reagent (*Drosophila melanogaster*) | UAS > stop > dTrpAmyrc | Bloomington Stock Center | # 66871 | |
| Genetic reagent (*Drosophila melanogaster*) | R71G01-GAL4 | Bloomington Stock Center | # 39599 | |
| Genetic reagent (*Drosophila melanogaster*) | R71G01-LexA | Bloomington Stock Center | # 54733 | |
| Genetic reagent (*Drosophila melanogaster*) | +; UAS-syteGFP, UAS-Denmark; Sb/+ | Rao Lab, Peking University | N/A | |
| Genetic reagent (*Drosophila melanogaster*) | UAS > stop > Kir2.1$^{eGFP}$ | Rao Lab, Peking University | N/A | |
| Genetic reagent (*Drosophila melanogaster*) | DskFlp | Pan Lab, Southeast University | N/A | |
| Genetic reagent (*Drosophila melanogaster*) | elav-GS | Zhong Lab, Tsinghua University | N/A | |
| Genetic reagent (*Drosophila melanogaster*) | UAS-dTrpA1/cyo | Garrity Lab, Brandeis University | N/A | |
| Genetic reagent (*Drosophila melanogaster*) | UAS-TNT | O'Kane Lab, University of Cambridge | N/A | |
| Genetic reagent (*Drosophila melanogaster*) | UAS-impTNT | O'Kane Lab, University of Cambridge | N/A | |
| Genetic reagent (*Drosophila melanogaster*) | UAS-Kir2.1 | Bloomington Stock Center | #6595, #6596 | |
| Genetic reagent (*Drosophila melanogaster*) | Dsk$^{GAL4}$ | Rao Lab, Peking University | N/A | |
| Genetic reagent (*Drosophila melanogaster*) | elav-GAL4 | Rao Lab, Peking University | N/A | |
| Genetic reagent (*Drosophila melanogaster*) | GluRIA$^{GAL4}$ | Rao Lab, Peking University | N/A | |
| Genetic reagent (*Drosophila melanogaster*) | CCKLR-17D3$^{GAL4}$ | Rao Lab, Peking University | N/A | |
| Genetic reagent (*Drosophila melanogaster*) | ΔDsk | Rao Lab, Peking University | N/A | |
| Genetic reagent (*Drosophila melanogaster*) | UAS > stop > myr::GFP | Gerald Rubin, Janelia Farm Research Campus | N/A | |
| Genetic reagent (*Drosophila melanogaster*) | ΔCCKLR-17D3 | Rao Lab, Peking University | N/A | |

*Continued on next page*

*Continued*

| Reagent type (species) or resource | Designation | Source or reference | Identifiers | Additional information |
|---|---|---|---|---|
| Genetic reagent (*Drosophila melanogaster*) | *ΔCCKLR-17D1* | Rao Lab, Peking University | N/A | |
| Genetic reagent (*Drosophila melanogaster*) | *UAS-Dsk* | Zhou Lab, Chinese Academy of Sciences, this paper | N/A | |
| Genetic reagent (*Drosophila melanogaster*) | *UAS-DskRNAi* | Pan Lab, Southeast University | N/A | |
| Genetic reagent (*Drosophila melanogaster*) | *elav-GAL4; UAS-dcr2* | Rao Lab, Peking University | N/A | |
| Genetic reagent (*Drosophila melanogaster*) | *Lexo-CD4-spGFP11/CyO*; *UAS-CD4-spGFP1-10/Tb* | **Gordon and Scott, 2009** | N/A | |
| Genetic reagent (*Drosophila melanogaster*) | *pC1-ss1* | Kaiyu Wang's lab, Institute of Neuroscience | N/A | |
| Genetic reagent (*Drosophila melanogaster*) | *pC1-ss2* | Kaiyu Wang's lab, Institute of Neuroscience | N/A | |
| Genetic reagent (*Drosophila melanogaster*) | *Dilp2-GAL4* | Zhong Lab, Tsinghua University | N/A | |
| Genetic reagent (*Drosophila melanogaster*) | *UAS-CCKLR-17D3* | Zhou Lab, Chinese Academy of Sciences, this paper | N/A | |
| Genetic reagent (*Drosophila melanogaster*) | *UAS-CCKLR-17D1RNAi* | Bloomington Stock Center | # 27494 | |
| Genetic reagent (*Drosophila melanogaster*) | *UAS-CCKLR-17D3RNAi* | Bloomington Stock Center | # 28333 | |
| Software, algorithm | MATLAB | MathWorks, Natick, MA | https://www.mathworks.com/products/matlab.html | |
| Software, algorithm | ImageJ | National Institutes of Health | https://imagej.nih.gov/ij/ | |
| Software, algorithm | Prism 7 | GraphPad | https://www.graphpad.com/ | |

## Fly stocks

Flies were reared on standard cornmeal-yeast medium under a 12 hr:12 hr dark:light cycle at 25°C and 60% humidity. Flies carrying a *dTrpA1* transgene were raised at 21°C. *UAS-TNTE* and *UAS-impTNT* were kindly provided by Dr Cahir O'Kane (University of Cambridge). *UAS-dTRPA1* was a gift from Dr Paul Garrity (Brandeis University). *Dilp2-GAL4* line, *trans*-Tango line, and *elav-GS* line were provided by Dr Yi Zhong (Tsinghua University), *DskFlp* and *DskRNAi* lines were provided by Dr Yufeng Pan (Southeast University). *UAS > stop > myr::GFP* (pJFRC41 in attP5) was a gift from Gerald Rubin, *UAS > stop > kir^eGFP^* was provided by Dr Yi Rao, *pC1-ss1* and *pC1-ss2* were provided by Dr Kaiyu Wang. The following lines were obtained from the Bloomington *Drosophila* Stock Center: *R71G01-GAL4* (BL#39599), *R71G01-LexA* (BL#54733), *TβH-GAL4* (BL#45904), TRIC line (BL#61679), *UAS-Kir2.1* (BL#6595 and BL#6596), *UAS-mCD8::GFP* (BL#5137), *UAS > stop > dTrpA^myrc^* (BL#66871). *Lexo-CD4-spGFP11/CyO*; *UAS-CD4-spGFP1-10/Tb* was previously described (**Gordon and Scott, 2009**).

## Behavioral assays

Flies were reared at 25°C. Virgin females and wild-type males were collected upon eclosion, placed in groups of 12 flies each and aged 5–7 days at 25°C and 60% humidity before carrying out behavior assay except for the thermogenetic experiments.

In female sexual behavior experiment in virgin females, mating behavior assays were carried out in the courtship chamber. A virgin female of defined genotype and a wild-type male were gently cold anesthetized and respectively introduced into two layers of the round courtship chambers separated

by a removable transparent film. The flies were allowed to recover for at least 1 hr before the film was removed to allow the pair of a test female and a wild-type male to contact. The mating behavior was recorded using a camera (Canon VIXIA HF R500) for 30 min at 30 fps for further analysis.

For female sexual behavior experiment in very young virgin females, we collected flies with 0–3 hr post-eclosion and measured receptivity at 12–16 hr post-eclosion using the same method as mentioned above.

For female sexual behavior experiment in mated females, we first collected virgin females upon eclosion and generated mated females by pairing females aged 5–7 days with wild-type males. Mated females were isolated for 18–24 hr and then assayed for receptivity with a new wild-type male using the same method as mentioned above.

For *dTrpA1* activation experiment, flies were reared at 21°C. Flies were loaded into courtship chamber and recovered for at least 30 min, then were placed at 21°C (control group) or 29°C (experimental group) for 30 min prior to removing the film and videotaping.

For egg laying experiment, virgin females were collected upon eclosion and five flies were housed on standard medium in single vials. The flies were transferred into new food tubes every 24 hr after aged 4 days, and we counted manually the number of eggs in each food tube.

For rejection behavior, the indicated genotype of virgin female paired with male, videotaped for 10 min at higher magnification, and scored manually for ovipositor extrusions.

For locomotor behavior experiment, virgin females were collected upon eclosion and placed in groups of 12 flies each. Individual females aged 5–7 days were used to test locomotor behavior, which was analyzed via Ctrax software (*Branson et al., 2009*).

## Immunohistochemistry

Whole brains of flies aged 5–7 days were dissected in 1× PBS and fixed in 2% paraformaldehyde for 55 min at room temperature. The samples were blocked in 5% normal goat serum for 1 hr at room temperature after washing the samples with PBT (1× PBS containing 0.3% Triton X-100) for four times for 15 min. Then, the samples were incubated in primary antibodies (diluted in blocking solution) for 18–24 hr at 4°C. Samples were washed four times with 0.3% PBT for 15 min, then were incubated in secondary antibodies (diluted in blocking solution) for 18–24 hr at 4°C. Samples were washed four times with 0.3% PBT for 15 min, then were fixed in 4% paraformaldehyde for 4 hr at room temperature. Finally, brains were mounted on poly-L-lysine-coated coverslip in 1× PBS. The coverslip was dipped for 5 min with ethanol of 30%→50%→70%→95%→100% sequentially at room temperature, and then dipped three times for 5 min with xylene. The brains were mounted with DPX and allowed DPX to dry for 2 days before imaging. Confocal images were obtained with Carl Zeiss (LSM710) confocal microscopes and Fiji software was used to process images. Primary antibodies used were: chicken anti-GFP (1:1000; Life Technologies), rabbit anti-DSK antibody (1:1000), mouse anti-nc82 (1:50; DSHB), rat anti-HA (1:100; Roche), mouse anti-GFP-20 (1:100; Sigma). Secondary antibodies used were: Alexa Fluor goat anti-chicken 488 (1:500; Life Technologies), Alexa Fluor goat anti-rabbit 546 (1:500; Life Technologies), Alexa Fluor goat anti-mouse 647 (1:500; Life Technologies), Alexa Fluor goat anti-rat 546 (1:500; Invitrogen) and Alexa Fluor goat anti-mouse 488 (1:500; Life Technologies).

## Generation of anti-DSK antibody

Rabbit anti-DSK antibody was generated previously (*Wu et al., 2020*). In brief, the anti-DSK antibody was generated by using the synthetic peptide N'-GGDDQFDDYGHMRFG-C' as antigen. The synthesis of antigen peptide, the production and purification of antiserum were performed by Beijing Genomics Institute (BGI).

## Generation of *UAS-Dsk* and *UAS-CCKLR-17D3*

*UAS-Dsk* was generated previously (*Wu et al., 2020*). In brief, *UAS-Dsk* constructs were injected and integrated into the attP40 site on the second chromosome through phiC31-mediated gene integration. The method of generation of *UAS-CCKLR-17D3* was same as described previously (*Wu et al., 2020*). Primer sequences for cloning the cDNA of *UAS-CCKLR-17D3* are as follows:

UAS-CCKLR-17D3

Forward:
ATTCTTATCCTTTACTTCAGGCGGCCGCAAAATGTTCAACTACGAGGAGGG

Reverse:
GTTATTTTAAAAACGATTCATTCTAGATTAGAGCTGAGGACTGTTGACG

## Genomic DNA extraction and RT-PCR

Genomic DNA was extracted from whole fly body using MightyPrep reagent for DNA (Takara). Whole head RNA was extracted from 50 fly heads using TRIzol (Ambion #15596018). cDNA was generated from total RNA using the Prime Script reagent kit (Takara).

## Validation of Δ*CCKLR-17D3*

Candidates of Δ*CCKLR-17D3* were characterized by the loss of DNA band in the deleted areas by PCR on the genomic DNA, as shown in *Figure 4A*. Primer sequences used for regions 1–4 in *Figure 4B* are as follows:

Region (1): Forward 5'- CAGTAGAGGATTCGCCTCCAAG-3'
Reverse 5'- GACATACAGCGAGAGTGC-3'
Region (2): Forward 5'- CATGAACGCCAGCTTCCG-3'
Reverse 5'- GCACTATTGGTGGTCACCAC-3'
Region (3): Forward 5'- GGAAATCATCTAACAGGCTTAC-3'
Reverse 5'- GCCGTGTCAAATCGCTTTC-3'
Region (4): Forward 5'- GCATACATACAAGCAAATTATGC-3'
Reverse 5'- CTCATATTCTTTTGGGCTACCAC-3'

Primer sequences used for amplifying *CCKLR-17D3* or *CG6891* cDNA in *Figure 4C* are as follows:

*CCKLR-17D3* cDNA: Forward 5'- GCCCATAGCGGTCTTTAGTC-3'
Reverse 5'- GTGATGAGGATGTAGGCCAC 3
*CG6891* cDNA: Forward 5'-GCTGTGTTCTGGATGTGGATG-3'
Reverse 5'- CTGGAACTGTGCTGGTTCTG-3'

## Drug feeding

Virgin females of defined genotype were collected upon eclosion and reared on standard cornmeal-yeast medium as a group of 12 for 4 days. Then, we transferred the female flies to new standard cornmeal-yeast food tube containing 500 µM RU486 (RU486+) or control solution (RU486-) for 2 days before behavior assay. RU486 (mifepristone; Sigma) was dissolved in ethanol.

## TRIC analysis

*Dsk*GAL4 flies were crossed with a TRIC line to detect the changes of intracellular $Ca^{2+}$ levels between virgin and mated females. Brains of virgin and mated females (2 days after copulation) were dissected and fixed with 8% paraformaldehyde for 2 hr, and then mounted with DPX. All the confocal images were obtained with Carl Zeiss (LSM710) confocal microscopes with the same settings.

Fiji software was used to process images. We first generated a Z stack of the sum of fluorescence signals, and then quantified the fluorescence intensity of DSK cell bodies of virgin and mated female brain, respectively. We quantified the TRIC signal by calculating the ratio of intensities of GFP signal over the RFP signal.

## Electrophysiological recordings

Young adult flies (1–2 days after eclosion) were anesthetized on ice and brain was dissected in saline solution. And the brain was continuously perfused with saline bubbled with 95% $O_2$/5% $CO_2$ (~pH 7.3) at room temperature. The saline composed of the following (in mM): 103 NaCl, 3 KCl, 4 $MgCl_2$, 1.5 $CaCl_2$, 26 $NaHCO_3$, 1 $NaH_2PO_4$, 5 *N*-tri-(hydroxymethyl)-methyl-2-aminoethane-sulfonic acid (TES), 20 D-glucose, 17 sucrose, and 5 trehalose.

Electrophysiological recordings were performed using a Nikon microscope with a 60× water immersion objective to locate target neurons. Then, we used Nikon A1*R*+ confocal microscope with infrared-differential interference contrast optics to visual for patch-clamp recordings and the image was shown on monitor by IR-CCD (DAGE-MTI). The recording pipette (~10–15 MΩ) was filled with internal solution containing 150 mg/ml amphotericin B. The internal solution consists of (in mM): 140 K-gluconate, 6 NaCl, 2 $MgCl_2$, 0.1 $CaCl_2$, 1 EGTA, 10 HEPES (pH 7.3). Current and voltage signals

were amplified with MultiClamp 700B, digitized with Digidata 1440A, recorded with Clampex 10.6 (all from Molecular Devices), filtered at 2 kHz, and sampled at 5 kHz. The recorded neuron was voltage clamped at –70 mV. Measured voltages were corrected for a liquid junction potential.

## Chemogenetic stimulation

ATP-gated ion channel $P2X_2$ was driven by *71G01-GAL4*. ATP-Na (Sigma-Aldrich) of 2.5 mM was delivered through a three-barrel tube (with the tip positioned ~50 μm away from the brain), controlled by stepper (SF77B, Warner Instruments) driven by Axon Digidata 1440A analog voltage output, allowing for fast solution change between perfusion saline and ATP stimulation.

## Brain image registration

A standard brain was generated using CMTK software as described previously (*Rohlfing and Maurer, 2003*; *Zhou et al., 2014*). Confocal stacks were then registered into the common standard brain with a Fiji graphical user interface as described previously (*Jefferis et al., 2007*).

## Connectomics analysis

The recently generated FAFB EM image set was used to identify the synaptic connections between pC1 neurons and DSK neurons (*Scheffer et al., 2020*). We got the number of synaptic connections and the unique identifier (Cell ID) from the following website: https://neuprint.janelia.org.

## Quantification and statistical analysis of female mating behavior

Two parameters including copulation rate and latency were used to characterize receptivity and we got the data sets of two parameters from same flies. The time from removing the film to copulation was measured for each female. The number of females that had engaged in copulation by the end of each 1 min interval were summed within 30 min and plotted as a percentage of total females for each time point. The time from removing the film to successful copulation for each female was used to characterize latency to copulation. And all the time points that female successfully copulated were analyzed by manual method and unhealthy flies were discarded. Three scorers with blinding to the genotypes and condition of the experiment were assigned for independent scoring.

## Statistical analysis

Statistical analyses were carried about using R software version 3.4.3 or GraphPad software. For the copulation rate, chi-square test is applied. For the latency to copulation, Kruskal-Wallis ANOVA test followed by post hoc Mann-Whitney U test was used for comparison among multiple groups. The Mann-Whitney U test was applied for analyzing the significance of two columns.

## Acknowledgements

We thank Yi Rao (Peking University), Yi Zhong (Tsinghua University), Cahir O'Kane (University of Cambridge), and Paul Garrity (Brandeis University) for providing fly lines. We thank Pengxiang Wu (Chinese Academy of Sciences), Yufeng Pan (Southeast University), Yinxue Wang (Max Planck Florida Institute for Neuroscience), and Yu Mu (Institute of neuroscience) for comments on the manuscript. This work is supported by grants to Chuan Zhou from National Natural Science Foundation of China (NO.Y711241133) and Strategic Priority Research Program of the Chinese Academy of Science (NO. Y929731103) and State Key Laboratory of Integrated Management of Pest Insects and Rodents, IOZ, CAS (NO.Y652751E03).

## Additional information

### Funding

| Funder | Grant reference number | Author |
| --- | --- | --- |
| National Natural Science Foundation of China | Y711241133 | Chuan Zhou |

| Funder | Grant reference number | Author |
|---|---|---|
| Chinese Academy of Sciences | Y929731103 | Chuan Zhou |

The funders had no role in study design, data collection and interpretation, or the decision to submit the work for publication.

## Author contributions

Tao Wang, Data curation, Formal analysis, Methodology, Resources, Writing – original draft; Biyang Jing, Formal analysis, Methodology, Software; Bowen Deng, Formal analysis, Investigation, Methodology; Kai Shi, Software; Jing Li, Data curation, Software; Baoxu Ma, Data curation; Fengming Wu, Formal analysis, Methodology, Writing – review and editing; Chuan Zhou, Conceptualization, Funding acquisition, Project administration, Supervision, Writing – review and editing

## Author ORCIDs

Chuan Zhou http://orcid.org/0000-0001-7952-7048

## Decision letter and Author response

Decision letter https://doi.org/10.7554/eLife.76025.sa1
Author response https://doi.org/10.7554/eLife.76025.sa2

## Additional files

### Supplementary files

• Supplementary file 1. Activation of Drosulfakinin (DSK) neurons did not change receptivity in mated females and very young females. The numbers were shown in the table are the number of pairs that copulated within the 30 min divided by the number of total tested pairs. None of females successfully copulated in mated females and very young females, except one mated female successfully copulated with the genotype of *UAS-dTrpA1/+* at 29°C.

• Supplementary file 2. Synaptic connections identified by electron microscopic (EM) reconstruction.

• Transparent reporting form

### Data availability

All data generated or analysed during this study are included in the manuscript and supporting file; source data files have been provided for all figures and figure supplements.

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
