## [Editor Report]

The manuscript by Wang and colleagues expands our understanding of the neural circuit mechanisms underpinning innate sexual behaviors in *Drosophila*. It exploits an arsenal of sophisticated tools to demonstrate that the neuropeptide Drosulfakinin (DSK) modulates female sexual receptivity via pC1-DSK-MP1-CCKLR-17D3 receptor expressing neurons. The study also introduces new transgenic tools that will be valuable for the community and will be of interest to neuroscientists exploring neuropeptide function and female sexual behavior.

---

## [Decision Letter]

**Decision letter after peer review:**

Thank you for submitting your article "Drosulfakinin signaling modulates female sexual receptivity in *Drosophila*" for consideration by *eLife*. Your article has been reviewed by 3 peer reviewers, including Sonia Sen as the Reviewing Editor and Reviewer #1, and the evaluation has been overseen by K VijayRaghavan as the Senior Editor.

Essential revisions:

1. Related to the writing:

- Introduction: We would like the authors to cover the appropriate literature (and references) related to (1) Dsk, (2) courtship ritual and its relevance to mate selection, and (3) female receptivity and a description of behaviours that constitute it.

- Results: We're recommending that the authors motivate each result section with (1) a rationale for the experiments done in that section, (2) a description of the experimental set-up, and (3) elaborate on the description and interpretation of the results for that section.

2. Statistics: We're concerned about the statistics used. For example, a non-parametric test (Χ2) has been used to analyze latency to copulation (measured in minutes). We're requesting that the authors elaborate on the statistics used in each of the results, and mention them in the legends. Specific points related to the statistics are in the detailed reviews below, but are also listed here:

- Copulation rate and latency to copulation should be presented as different data sets and statistically analyzed independently.

- The y scale of the same type of graphs should be the same across figures. The scale is truncated in many graphs showing latency to copulation. This should be fixed.

- The authors should state which experimental groups are compared in each figure. When more than two genotypes are plotted, multiple comparison tests should be applied (e.g., see Figure 2D, latency to copulation; Figure 5D 5F, latency to copulation; Figure 6E, latency to copulation). All comparisons should be reported, either in each graph or in a supp. table.

- In some figure legends the authors state 'Kruskal-Wallis and post-hoc Mann-Whitney U tests or post-hoc Student's T-test'. For example, in Figure 1-suppl 3. The authors should clearly indicate which test is used to analyze each data set.

- What's the rationale for using a post-hoc Student's T-test in Figure 1-suppl 2? A Conover post hoc test might be more appropriate.

3. Related to the Dsk and receptor tools used.

- We would like the authors to show the expression pattern of the Dsk lines used:

- Expression pattern of the whole Gal4 in the VNC and to comment on its expression in non-neuronal tissues.

- Expression pattern of the split lines for the Dsk-m and Dsk-l clusters.

- Verify the RNAi lines used.

- Expression pattern of the receptor-line used (CCKLR-17D3GAL4).

4. In some places, it's ambiguous what tools were generated in this study vs their older paper. Can the authors please state this unambiguously for all tools and reagents used in this manuscript?

5. We're recommending that the authors comment on what aspects of the female sexual behaviour is altered in these manipulations? We presume they have videos that they can examine for this.

6. Could the authors clarify which tracking tools were used for locomotor assessment? We have concerns that Ctrax cannot maintain the identity of two or more flies in an arena and recommend using another tracking tool for two fly assays.

While these are the revisions we recommend as essential, we were wondering if the authors might already have tried an alternative Dsk-Gal4 that targets the IPC cluster? If they have, it would be nice to include it here. If not, could they please discuss the possibility that this cluster might also be involved sexual receptivity.

Similarly, if the authors have already investigated the requirement of the CCKLR-17D3GAL4 neurons in sexual receptivity behaviour it would be nice to include in this manuscript.

However, if these experiments have not already been done, or difficult to do, given the current situation, we are not listing them as essential.

*Reviewer #1 (Recommendations for the authors):*

I have a few suggestions to the authors that might make this data more accessible to the reader.

- Introduction: As it's currently written, the introduction doesn't equip the reader to interpret the results in the context of what's known in the field. Could the authors please revisit this section and elaborate (to the extent relevant to this manuscript) on the Dsk literature, male courtship and sexual receptivity in females?

- Writing in the Results section:

- As the authors present their data, it is sometimes unclear what the motivation behind the experiment was. For example, why did they look at egg-laying behaviour (line 96), or TRIC (line 97) (do please also elaborate on what TRIC is). Could the authors please pay attention to this?

- I'd urge the authors to briefly explain the set up of each experiment where the relevant data is being presented. This helps the reader tune into the experiment and its interpretation without having to go to the methods section.

- The circuit related experiments: The authors show evidence for 71G01-neurons > Dsk-m-neurons > CCKLR-17D3 receptor expressing neurons (not identified). I would have liked to have seen a more refined analysis of circuity within these domains. So, I had a few comments/questions regarding this section:

- 71G01-neurons: This line has been used extensively in literature – including by the authors – to access the P1 neurons in males. I believe it also labels the pC1 neurons in females. Are the authors suggesting that the pC1 neurons are the upstream neurons to Dsk-m? This would be interesting.

- The CCKLR-17D3-neurons: I believe the authors have a knock-in Gal4 of this receptor that is not too busy. I would be keen to know if any one of these neurons are downstream of the Dsk neurons.

*Reviewer #2 (Recommendations for the authors):*

Experimental procedures should include details about the methods and genetic tools used in the manuscript e.g., origin of Dsk-GAL4 and Dsk antibodies.

Please provide the expression pattern on the Dsk-Gal4 used in CNS. Does it cover the π region? If not, the experiments involving dissection of Dsk function should be repeated using an additional Dsk-GAL4 driver that includes the Dsk neurons from IPC (Figure 1, & 2), this would be required to strengthen the manuscript about involvement of Dsk neurons (not a subset) in female receptivity.

Please mention in the main text that Dsk antibodies used in Figure 1D do not label the full set of Dsk neurons. Please also cite earlier literature that shows these neurons are covered by Dsk antibodies.

What are various UAS-Dsk and Dsk deletion lines mentioned in Figure 1 E-J?

For experiments involving dTRPA1, higher temperature might affect female behavior. Authors may consider using various optogenetic tools to dissect these behaviors in future. A detailed video analysis of various genotypes should be done to show that locomotion is normal, and describe other sexual behaviors beside copulation rate and latency, please also include this source data.

Genetic background has a significant impact on behavior and it's not clear if outcrossing was performed in the same genetic background for various fly stocks used in the manuscript? If not, outcrossing should be performed for at least 5-7 generations for behavioral experiments in Figure 1-2, to rule out effects of genotype.

Figure 1 Supplement 1, provide details for the Dsk-RNAi used and what is the effect without using dicer 2 as it can lead to off targets? What is the phenotype when the RNAi is driven by an appropriate Dsk-GAL4?

Was the locomotion analysis done in the presence of males (Figure 1, Supplement 2 & Figure 6, Supplement 3)? While the discussion mentions use of Ctrax for locomotion tracking, it is not mentioned anywhere in the main text of experimental procedures. Also, Ctrax lacks the ability to faithfully maintain the identity of two or more flies in an arena. It would be better to use another tracking tool for two fly assays and *Drosophila* activity monitors for single fly locomotion assays.

Figure 1, Supplement 4, please include source data for TRIC experiments.

GRASP data (Figure3, Supplement 2) is weak and does not show much overlap between Dsk and R70G01GAL4 when compared to trans Tango experiments. These experiments should be either repeated or the caveat should be mentioned in the manuscript.

Figure 5E- the curve is really skewed towards the bottom as compared to other genotypes, are these controls healthy? It might be better to repeat this experiment/ provide video analysis.

There are several grammatical errors and awkward sentence constructions throughout the manuscript, please correct this. Some examples are below-

Introduction section needs substantial re-writing and the study needs to be properly placed in the broader context of the field, see my comments above for appropriate citations, below are a few suggestions but this is not an exhaustive list.

Line 48-50, Spinster mutants actively reject males, and it is not a case of 'lower mating success rate' as females are rejecting males.

Line 71-72, [missing citation] including satiety/food ingestion- cite Soderberg et al., 2012.

Line 72-73, [missing citation] aggression- cite Williams et al., Genetics 2014 and Agrawal et al., JEB 2020.

Wu et al., Nature Communications 2019 have extensively characterized role of Dsk in male courtship and sexual drive and this work is not cited properly.

Line 104, correct Figure 2I to Figure 1I and line 108, instead of Figure 2J correct to 1J.

Line 132 introduces R71G01-GAL4 neurons awkwardly and it seems to come out of nowhere. What was the role of that driver? Why did the authors choose to investigate this GAL4 driver?

Line 175 mentions GluRIA-GAL4 but the reference is missing, please also include the role of this driver.

Line 191-192, states- 'we constructed knockout lines for these two receptors (figure 6A-C)' but in the figure only one receptor is shown. Are these the same lines as in the earlier study (Wu et al., 2020)?

Line 194-196, RNAi knockdown of CCKLR-17D3 was performed, what is the phenotype from CCKLR-17D1 knock-down?

Line 239, After Dsk- insert 'gene'.

Line 267, classification of Dsk neurons, given that Wu et al., Nat. Comm., 2019 study precedes these studies, classification of Dsk neurons and any differences should be mentioned in the main text itself rather than in the Discussion section.

Line 270, for male aggression please also cite Agrawal et al., JEB 2020.

line 488, by RCR do you mean PCR?

Figure supplements (e.g., Figure 1 figure supplement 6,) repeatedly show the behavioral arena used for courtship assay. This is redundant and should be shown only once and referred to throughout the manuscript as required.

*Reviewer #3 (Recommendations for the authors):*

I believe the science and its presentation could be strengthened by addressing the points below:

Science.

1. It is not clear whether the authors have used proper statistical tests. The same flies are monitored over time for copulation rate, which should be considered in the statistical analysis.

2. Why have the authors applied a non-parametric test (Χ2) for analyzing latency to copulation (measured in minutes)?

3. Copulation rate and latency to copulation should be presented as different data sets and statistically analyzed independently.

4. The y scale of the same type of graphs should be the same across figures.

5. The scale is truncated in many graphs showing latency to copulation. This should be fixed.

6. The authors should state which experimental groups are compared in each figure. When more than two genotypes are plotted, multiple comparison tests should be applied (e.g., see Figure 2D, latency to copulation; Figure 5D 5F, latency to copulation; Figure 6E, latency to copulation). All comparisons should be reported, either in each graph or in a supp. table.

7. In some figure legends the authors state 'Kruskal-Wallis and post-hoc Mann-Whitney U tests or post-hoc Student's T-test'. For example, in Figure 1-suppl 3. The authors should clearly indicate which test is used to analyze each data set.

8. What's the rationale for using a post-hoc Student's T-test in Figure 1-suppl 2? A Conover post hoc test might be more appropriate.

9. The data strongly support a role for DSK and DSK neurons control female sexual receptivity. However, it is unclear whether DSK neurons in the VNC contribute to the observed phenotypes. An experiment addressing this point would further support the findings of the study.

10. Figure 1 A—figure supplement 4. None of the control virgin females have laid any unfertilized eggs across 3 days, which is unexpected.

11. Figure 1B-suppl 4. More details should be added in the graph and legend to understand the TRIC data.

12. Do the authors have data showing which aspects of female sexual behavior are regulated by the DSK signaling? For example, does activation of DSK neurons induce any female acceptance behaviors (e.g., partial ovipositor extrusion)? Perhaps DSK neurons trigger motor outputs associated with sexual receptivity? If DSK neurons acted downstream of pC1 neurons, we would expect them to induce vaginal plate opening. These possibilities should be experimentally addressed or discussed in the manuscript.

13. I believe more information about the expression pattern of Dsk in neuronal and non-neuronal tissues would help better understand the findings. Details of the number of DSK neurons, different brain clusters, as well as projection patterns, would help the authors to put their findings in context. The authors should add confocal images showing the expression pattern of Dsk neurons in VNC.

A better description of the expression pattern of Dsk would improve the narrative and help the authors better explain (1) the anatomical and functional experiments linking DSK neurons with 71G01-Gal4 neurons (2) the identification of subsets of DSK clusters key for sexual receptivity (3) the importance of manipulating DSK and Dsk-expressing cells solely in the nervous system.

14. *Dsx*-expressing neurons have a prominent role in controlling female sexual receptivity. Is there any indication that DSK neurons and *Dsx* neurons fully/partially overlap?

15. I believe Table 1 is missing in the manuscript.

16. Some genetic controls are missing in several figures. For instance, in Figure 2, elav-GAL4/+ is missing.

17. The authors should clarify that the GRASP experiments don't provide information about the direction of connectivity.

18. The authors should add images showing the pattern of DSK neurons targeted with genetic intersections using GluRIA-Gal4 and TbH-Gal4 with DSK-FLP in the VNC.

Presentation:

I believe a substantial improvement in the writing style would help readers fully judge and appreciate the findings of this interesting study.

1. In my opinion, the opening paragraph of Introduction is not engaging. Overall, I think the authors could strengthen the Introduction by adding information about the courtship ritual in flies and explaining its relevance for mate selection. Moreover, they could describe how female flies signal sexual receptivity and accept a male for copulation. This would help them highlight the importance of their work and make it more attractive for non-specialist readers.

2. My impression is that the Results section reads like a long list of results, which are not connected to tell an interesting story. The rationale of many of the experiments are either presented in the Discussion section or not discussed at all. The authors could speculate how DSK neurons/ DSK might work to regulate sexual receptivity to introduce subsequent experiments in the main text. A better description of the results and their potential implications would help the narrative of the study (more details below).

3. One of the strengths of this study is the arsenal of modern tools used to support the findings. However, many of the methods are not described in the main text. More information would help readers to fully grasp the approaches and findings of this study.

4. The Discussion section superficially discusses the findings of the study. It rather reads like a recapitulation of the results. The authors could enrich the discussion by speculating how DSK neurons might control female sexual receptivity. Could pC1 neurons (targeted by 71G01-Gal4?) act as a central node for female sexual receptivity, conveying female mating decisions to Dsk neurons? Alternatively, could DSK neurons also integrate sensory information relevant to sexual behavior? Are there any sensory pathways relevant to female sexual receptivity feeding into DSK neurons? What are the potential downstream targets of DSK? What aspects of female sexual behavior are regulated by the DSK pathway? Do females show increased vaginal plate/ partial ovipositor extrusion? How do DSK neurons differ between males and females, and how do these differences may result in different sex-specific behaviors?

5. The schematic in Figure 4 A is not clear.

6. Figure 4. A bigger schematic of the brain depicting the different DSK clusters would help understand which neurons are assessed.

7. The authors should make the figures accessible to readers with color-blindness.

8. I suggest the authors proofread the manuscript, as there are typos, grammatical mistakes and unclear sentences throughout the manuscript. Some examples:

Line 46 – 'is' should be changed to 'are'.

Line 87 – This sentence contains grammatical mistakes.

Line 93 – RNAi-mediated females: this should be better defined.

Line 109 – 'Indicated' should be changed to 'indicate'.

Line 104 – Figure is wrongly cited. It should be Figure 1I.

Line 112 – 'were' should be changed to 'are'.

Line 132 – 'expressed' should be changed to 'are present'.

Line 134 -Fix this sentence 'Given that, we hypothesized whether DSK'.

Line 139 – Change 'recombinant' to 'reconstituted'.

Line 157 -The sentence does not make sense.

Line 169 – Unclear sentence.

Line 188 – It should be-'wanted' not 'want'.

Line 270 – The sentence should be rewritten: In this study, DSK neurons also modulate female sexual behavior.

Line 488 – Fix this sentence 'RCR analysis from genomic DNA samples'

Line 502 – Fix this sentence: 'And there are same numbers in two parameters'.

'Latency' should be described as' latency to copulation'.

Figure 1.Suppl 5 – fix the typo in the title.

The authors should better articulate the rationale of the experiments and explain the techniques so readers can better appreciate the findings. For example:

Line 86 – 'behavior, we first constructed knock out line for Dsk'. Describe how the knock out line for DsK was made.

Line 94 – Explain the rationale for looking at male courtship behavior.

Line 96 – Explain the rationale for looking at egg laying.

Line 97 – Explain the rationale of looking at TRIC signal changes. The authors should also explain how this technique works and provide more details on the findings.

The authors should clearly define if any of the tools used in this study were previously generated.

Line 102 – Define the DsK Gal4 used in Figure 1I. Is it the same line used in subsequent Figures?

Line 109 – The conclusion should also state that the evidence points to a role of DSK neurons in female sexual receptivity.

Line 119 – Explain what's the rationale for looking at very young females. State their age and differences with previous virgin females tested in other experiments.

Line 133 – More information about 71G01-Gal4, and the fact that it might intersect pC1 neurons, would help explain the rationale behind testing these neurons.

Line 157 – Here it is important to explain how many DSK neurons are present in the brain, and how they are distributed in different clusters (by including images of the nervous system).

Line 175 – The authors identify two Gal4 lines (GluRIAGAL4 and TβHGAL4) that help them intersect different DSK neurons but these lines are not described. The authors should explain what neurons are targeted by these drivers (e.g., glutamatergic, octopaminergic, etc). This information could be useful to interpret and discuss their findings (e.g., relevant DSK neurons might be excitatory).

Line 164 – Explain how the P2X2 optogenetic approach works.

Line 191 – 'We constructed knock out lines for these two receptors (Figure 6A-C) (Wu et al., 2020)'. Clearly state if these are novel tools created for this study.

---

## [Author Response]

Essential revisions:1. Related to the writing:- Introduction: We would like the authors to cover the appropriate literature (and references) related to (1) Dsk, (2) courtship ritual and its relevance to mate selection, and (3) female receptivity and a description of behaviours that constitute it.

We thank the reviewer for this comment, and thoroughly rewrote the introduction and added appropriate references as suggested.

- Results: We're recommending that the authors motivate each result section with (1) a rationale for the experiments done in that section, (2) a description of the experimental set-up, and (3) elaborate on the description and interpretation of the results for that section.

Thanks for the suggestions. In the revised manuscript, we have explained the rationale of experiments (such as egg-laying; TRIC; courtship behavior; female sexual behavior in very young virgin female and mated female) and described the experiment and results more accurately.

2. Statistics: We're concerned about the statistics used. For example, a non-parametric test (Χ2) has been used to analyze latency to copulation (measured in minutes). We're requesting that the authors elaborate on the statistics used in each of the results, and mention them in the legends. Specific points related to the statistics are in the detailed reviews below, but are also listed here:

Thanks for pointing this out. We have carefully checked statistics and described statistics in detail in each figure legend.

- Copulation rate and latency to copulation should be presented as different data sets and statistically analyzed independently.

In the revised manuscript, the data sets including copulation rate and latency to copulation are now presented independently. In addition, we have re-analyzed the latency to copulation using proper statistical test (For the copulation rate, chi-square test is applied. For the latency to copulation, Kruskal-Wallis and post-hoc Mann-Whitney U tests are applied).

- The y scale of the same type of graphs should be the same across figures. The scale is truncated in many graphs showing latency to copulation. This should be fixed.

We have corrected this issue throughout.

- The authors should state which experimental groups are compared in each figure. When more than two genotypes are plotted, multiple comparison tests should be applied (e.g., see Figure 2D, latency to copulation; Figure 5D 5F, latency to copulation; Figure 6E, latency to copulation). All comparisons should be reported, either in each graph or in a supp. table.

Thanks for pointing this out. In the revised manuscript, we have re-analyzed the latency to copulation using proper statistical test as mentioned above and revised the description in the figure legend.

- In some figure legends the authors state 'Kruskal-Wallis and post-hoc Mann-Whitney U tests or post-hoc Student's T-test'. For example, in Figure 1-suppl 3. The authors should clearly indicate which test is used to analyze each data set.

In the revised manuscript, we have carefully examined all figure legends about statistical test and clarified the statistical tests used for each data set.

- What's the rationale for using a post-hoc Student's T-test in Figure 1-suppl 2? A Conover post hoc test might be more appropriate.

We have corrected the statistics in the revised manuscript throughout.

3. Related to the Dsk and receptor tools used.- We would like the authors to show the expression pattern of the Dsk lines used:

In the revised manuscript, we have added the expression pattern of the Dsk^GAL4^ in the brain and the ventral nerve cord (please see Figure 1—figure supplement 3D).

- Expression pattern of the whole Gal4 in the VNC and to comment on its expression in non-neuronal tissues.

In the revised manuscript, we have added the expression pattern of the GAL4 lines in the VNC (*e.g.*, Dsk^GAL4^; please see Figure R1); and GAL4 lines labelling DSK-MP1 and DSK-MP3 neurons respectively (please see Figure 3—figure supplement 1). In addition, we also added the expression pattern of the Dsk^GAL4^ in the gut and no expression patterns were observed in the glia or gut (please see Figure 1—figure supplement 3D-E).

- Expression pattern of the split lines for the Dsk-m and Dsk-l clusters.

To be in accordance with previous studies, we now used the term DSK-MP1 for the middle DSK-M neurons and DSK-MP3 for the lateral DSK-L neurons in the revised manuscript. We show the expression pattern of the split lines for DSK-MP1 and DSK-MP3 clusters in Figure 3A-B (brain) and Figure 3—figure supplement 1 (VNC),.

- Verify the RNAi lines used.

The Dsk-RNAi line used in this study is a gift from Yufeng Pan’s lab (Wu et al. 2019, Nat. Comm.). We verify the efficiency of this RNAi by anti-DSK staining in females and found that anti-DSK signals are significantly decreased after knocking down the expression of Dsk gene (please see Figure 1—figure supplement 3A-B). Furthermore, we further performed RNAi interference experiment by using *Dsk^GAL4^* drive the expression of Dsk-RNAi to test the change in female sexual behavior (please see Figure 1—figure supplement 3F).

- Expression pattern of the receptor-line used (CCKLR-17D3GAL4).

In the revised manuscript, we have added the expression pattern of *CCKLR-17D3^GAL4^* in the brain and the ventral nerve cord (please see Figure 4—figure supplement 4A).

4. In some places, it's ambiguous what tools were generated in this study vs their older paper. Can the authors please state this unambiguously for all tools and reagents used in this manuscript?

Thanks for pointing this out. Indeed, a few reagents were generated in our previous *eLife* paper (Wu et al., 2020). In the revised manuscript, we clearly indicated whether the genetic reagent was generated in this study or a previous study.

5. We're recommending that the authors comment on what aspects of the female sexual behaviour is altered in these manipulations? We presume they have videos that they can examine for this.

Thanks for the suggestion. We re-recorded high resolution videos. We asked whether the phenotype of decreased female receptivity in *Dsk* mutant flies is due to potentially elevated ovipositor extrusion (a rejection behavior). However, we found that manipulating *Dsk* gene did not affect ovipositor extrusion (please see Figure 1—figure supplement 2). Similarly, we also analyzed whether activating DSK neurons would affect ovipositor extrusion in females with courting males. We found that manipulation of DSK neurons did not affect ovipositor extrusion (please see Figure 2—figure supplement 1). Finally, we further discussed the potential effect of Dsk signaling on female sexual behavior (*e.g.*, potential role of pausing behavior to male courtship) in the Discussion section.

6. Could the authors clarify which tracking tools were used for locomotor assessment? We have concerns that Ctrax cannot maintain the identity of two or more flies in an arena and recommend using another tracking tool for two fly assays.

Thanks for the suggestion. Indeed, we used single female in the absence of a courting male for the locomotor test and used Ctrax software for data analysis. We could not faithfully analyze locomotor behaviors in a pair of courting flies, and we believe locomotor in single flies could faithfully reflect their general locomotor activity.

While these are the revisions we recommend as essential, we were wondering if the authors might already have tried an alternative Dsk-Gal4 that targets the IPC cluster? If they have, it would be nice to include it here. If not, could they please discuss the possibility that this cluster might also be involved sexual receptivity.

We thank the reviewer for this comment. Indeed, our *Dsk^GAL4^* does not label the IPCs, and it has been found that DSK is also expressed in the IPCs. Although we could not obtain another Dsk-Gal4 that may target the IPCs, we used the Dilp2-GAL4 that specifically labels at least some Dsk-expressing IPCs. We found that restricting the expression of DskRNAi in IPC neurons using the Dilp2-GAL4 does not affect female sexual behavior (please see Figure 1—figure supplement 3G). These results indicate that Dsk peptide released from IPCs is not involved in regulating female sexual behavior.

Similarly, if the authors have already investigated the requirement of the CCKLR-17D3GAL4 neurons in sexual receptivity behaviour it would be nice to include in this manuscript.

Thanks for the suggestion. In the revised manuscript, we have examined the role of CCKLR-17D3 neurons in regulating female sexual behavior (please see Figure 4H-J) and modified the text accordingly.

However, if these experiments have not already been done, or difficult to do, given the current situation, we are not listing them as essential.Reviewer #1 (Recommendations for the authors):I have a few suggestions to the authors that might make this data more accessible to the reader.- Introduction: As it's currently written, the introduction doesn't equip the reader to interpret the results in the context of what's known in the field. Could the authors please revisit this section and elaborate (to the extent relevant to this manuscript) on the Dsk literature, male courtship and sexual receptivity in females?

We thank the reviewer for this suggestion. In this new version, we carefully rewrote the introduction by adding relevant information to make it more concisely.

- Writing in the Results section:- As the authors present their data, it is sometimes unclear what the motivation behind the experiment was. For example, why did they look at egg-laying behaviour (line 96), or TRIC (line 97) (do please also elaborate on what TRIC is). Could the authors please pay attention to this?

We thank the reviewer for pointing out this issue. We have carefully fixed the description by adding the motivation behind every experiment.

- I'd urge the authors to briefly explain the set up of each experiment where the relevant data is being presented. This helps the reader tune into the experiment and its interpretation without having to go to the methods section.

We have briefly explained the set-up of each experiment in the figure legends. For example, for the set-up in Figure 6A: “(A) Left: ATP stimulation and recording arrangement. The chemical stimulation is implemented using a three-barrel tube (with the tip positioned ~50μm away from the brain), controlled by a stepper for rapid solution change. Right: schematic illustrating the activation of *R71G01GAL4* neurons by ATP and patch-camp recording of DSK neurons.”

- The circuit related experiments: The authors show evidence for 71G01-neurons > Dsk-m-neurons > CCKLR-17D3 receptor expressing neurons (not identified). I would have liked to have seen a more refined analysis of circuity within these domains. So, I had a few comments/questions regarding this section:- 71G01-neurons: This line has been used extensively in literature – including by the authors – to access the P1 neurons in males. I believe it also labels the pC1 neurons in females. Are the authors suggesting that the pC1 neurons are the upstream neurons to Dsk-m? This would be interesting.

We thank the reviewer for this suggestion. We have revised the text more clearly. In the revised manuscript, we stated that R71G01-GAL4 labels the pC1 neurons as well as many other neurons in females, and speculate that there is a functional connection between pC1 neurons and DSK-MP1 neurons. In order to further confirm this speculation, we now requested two splitGAL4 driver lines that specifically label pC1 neurons, pC1-ss1 and pC1-ss2 (Wang et al., 2021, Nature) from Dr. Kaiyu Wang who recently set up a lab in Shanghai. Indeed, we found that pC1 neuron axons overlapped with DSK neuron dendrites by anatomical registration (please see Figure 5—figure supplement 4). Furthermore, we utilized the recently generated full adult female brain (FAFB) electron microscopic (EM) image set and found that pC1 neurons have intense synaptic input on DSK-MP1 neurons, especially the single pair of DSK-MP1b neurons (Table S2). We revised our text accordingly and further discussed this possibility in the discussion session of the revised manuscript.

- The CCKLR-17D3-neurons: I believe the authors have a knock-in Gal4 of this receptor that is not too busy. I would be keen to know if any one of these neurons are downstream of the Dsk neurons.

Yes, we have a knock-in GAL4 of CCKLR-17D3 and we also showed the expression pattern (please see Figure4—figure supplement4A). It is well known that *dsx* broadly expresses in female brain and plays a key role in regulating female sexual behavior. Thus, we asked whether *CCKLR-17D3^GAL4^* drives expression in *dsx* neurons to regulate female receptivity. However, intersection between *CCKLR-17D3^GAL4^* and *dsx^LexA^* only labeled projections from peripheral sensory neurons that innervate the SOG region (please see Figure 4—figure supplement 4B). Furthermore, either overexpressing or knocking down CCKLR-17D3 in all *dsx* neurons did not alter virgin female receptivity (please see Figure 4—figure supplement 4C-D). These results indicate that CCKLR-17D3 did not function in *dsx* neurons to regulate female sexual behavior. In the future, we need to generate *CCKLR-17D3-Flp* line to further subdivide CCKLR-17D3 neurons using intersectional technology and investigate which neuron clusters are responsible for female sexual behavior.

Reviewer #2 (Recommendations for the authors):Experimental procedures should include details about the methods and genetic tools used in the manuscript e.g., origin of Dsk-GAL4 and Dsk antibodies.

Indeed, most lines used in this study were the same with those used in our previous *eLife* paper (Wu et al., 2020). We have modified the text and also briefly explained the generation of these genetic tools in main text or method section.

Please provide the expression pattern on the Dsk-Gal4 used in CNS. Does it cover the π region? If not, the experiments involving dissection of Dsk function should be repeated using an additional Dsk-GAL4 driver that includes the Dsk neurons from IPC (Figure 1, & 2), this would be required to strengthen the manuscript about involvement of Dsk neurons (not a subset) in female receptivity.

We thank the reviewer for pointing out this issue and we have added the expression pattern of the Dsk^GAL4^ in the brain and ventral nerve cord (please see Figure 1—figure supplement 3D). Indeed, our DskGAL4 does not label the IPCs, and it has been found that DSK is also expressed in the IPCs. We do not test the role of DSK neurons driven by another Dsk-Gal4 driver, as we cannot obtain this Dsk-Gal4 that may target the IPCs. However, we examined the effects of knocking down the expression of *Dsk* only in insulin-producing cells (IPCs) in π region on female sexual behavior as shown above.

Please mention in the main text that Dsk antibodies used in Figure 1D do not label the full set of Dsk neurons. Please also cite earlier literature that shows these neurons are covered by Dsk antibodies.

We thank the reviewer for pointing out this and have modified the text as suggested as following: “…immunostaining with anti-DSK antibody, which does not label the full set of DSK neurons as previously found (Nichols and Lim, 1996).”.

What are various UAS-Dsk and Dsk deletion lines mentioned in Figure 1 E-J?

We thank the reviewer for pointing this out and we have changed the descriptions of these lines. We now stated clearly that these lines were generated in our previous study and also briefly described how they were generated in the main text and method section.

For experiments involving dTRPA1, higher temperature might affect female behavior. Authors may consider using various optogenetic tools to dissect these behaviors in future. A detailed video analysis of various genotypes should be done to show that locomotion is normal, and describe other sexual behaviors beside copulation rate and latency, please also include this source data.

As suggested by the reviewer, we have analyzed the general locomotion activity of dTrpA1 experiment. We found that females with activating of DSK neurons (*Dsk^GAL4^>UAS-dTrpA1*) did not affect locomotion behavior compared with genetic control (*Dsk^GAL4^/+ or UAS-dTrpA1/+*) in high or low temperature condition (please see Figure 2—figure supplement 2A). Our results demonstrated that higher temperature did not affect female sexual behavior (Figure 2B-C), although higher temperature might induce higher locomotion velocity. In addition, we also tested the general locomotion activity after inactivating DSK neurons and found that locomotion did not significant change compared with controls (please see Figure 2—figure supplement 2B).

As suggested by the reviewer, we have re-recorded high resolution videos to analyze whether activation of DSK neurons affect ovipositor extrusion of female with courting males. We found that manipulation of DSK neurons did not affect the phenotype of ovipositor extrusion (please see Figure 2—figure supplement 1).

Genetic background has a significant impact on behavior and it's not clear if outcrossing was performed in the same genetic background for various fly stocks used in the manuscript? If not, outcrossing should be performed for at least 5-7 generations for behavioral experiments in Figure 1-2, to rule out effects of genotype.

We understand that reviewer’s concern regarding the genetic background. Indeed, the flies used in this study were backcrossed to isogenized Canton S flies for at least five generations prior to behavior studies to eliminate the effect of genotype.

Figure 1 Supplement 1, provide details for the Dsk-RNAi used and what is the effect without using dicer 2 as it can lead to off targets? What is the phenotype when the RNAi is driven by an appropriate Dsk-GAL4?

The Dsk-RNAi line used in this study is a gift from Yufeng Pan’s lab (Wu et al. 2019, Nat. Comm.). We verify the efficiency of this RNAi by anti-DSK staining in females and found that anti-DSK signals are significantly decreased after knocking down the expression of *Dsk* gene (please see Figure 1—figure supplement 3A-B). We have performed RNAi interference experiment by using elav-GAL4 to drive the expression of Dsk-RNAi and found that knocking down the expression of *Dsk* significantly decreased copulation rate and prolonged the latency to copulation. In addition, we also found that knocking down the expression of *Dsk* by using Dsk^GAL4^ to drive UAS-DskRNAi significantly decreased copulation rate and prolonged the latency to copulation (please see Figure 1—figure supplement 3F).

Was the locomotion analysis done in the presence of males (Figure 1, Supplement 2 & Figure 6, Supplement 3)? While the discussion mentions use of Ctrax for locomotion tracking, it is not mentioned anywhere in the main text of experimental procedures. Also, Ctrax lacks the ability to faithfully maintain the identity of two or more flies in an arena. It would be better to use another tracking tool for two fly assays and *Drosophila* activity monitors for single fly locomotion assays.

Thanks for the suggestion. Indeed, we used single female in the absence of a courting male for the locomotor test and used Ctrax software for data analysis. We could not faithfully analyze locomotor behaviors in a pair of courting flies, and we believe locomotor in single flies could faithfully reflect their general locomotor activity.

Figure 1, Supplement 4, please include source data for TRIC experiments.

We have re-drawn this figure including source data as reviewer suggested (please see Figure 1—figure supplement 6B-D). We re-quantified the TRIC signal by calculating the ratio of intensities of GFP signal over the RFP signal.

GRASP data (Figure3, Supplement 2) is weak and does not show much overlap between Dsk and R70G01GAL4 when compared to trans Tango experiments. These experiments should be either repeated or the caveat should be mentioned in the manuscript.

As suggested by the reviewer, we have repeated the GRASP experiment (please see Figure 5-supplement 2).

Figure 5E- the curve is really skewed towards the bottom as compared to other genotypes, are these controls healthy? It might be better to repeat this experiment/ provide video analysis.

We understand reviewer’s concern regarding Figure 5E. All flies tested for behavior have no obvious developmental deficit. We now repeated this experiment and found similar results. These control females showed slightly reduced copulation rate compared with some other control females. which could be due to the complex genetic manipulations. Thus, we tested up to a hundred pairs of flies (in many cases, n > 100) for most experiments and provided as many control experiments as we could in this study.

There are several grammatical errors and awkward sentence constructions throughout the manuscript, please correct this. Some examples are below-Introduction section needs substantial re-writing and the study needs to be properly placed in the broader context of the field, see my comments above for appropriate citations, below are a few suggestions but this is not an exhaustive list.

We carefully rewrote the introduction part by adding the information related to this manuscript to make it more concise and we also have correctly cited these references as mentioned above in the revised manuscript.

Line 48-50, Spinster mutants actively reject males, and it is not a case of 'lower mating success rate' as females are rejecting males.

We thank the reviewer for pointing out this issue and we have changed the description of *spinster* gene in revised manuscript as following: “…mutant females of *spinster* show enhanced rejection behavior”.

Line 71-72, [missing citation] including satiety/food ingestion- cite Soderberg et al., 2012.

We thank the reviewer for pointing out this issue we have added this reference in the revised manuscript.

Line 72-73, [missing citation] aggression- cite Williams et al., Genetics 2014 and Agrawal et al., JEB 2020.

We thank the reviewer for pointing out this issue and we have added these references in the revised manuscript.

Wu et al., Nature Communications 2019 have extensively characterized role of Dsk in male courtship and sexual drive and this work is not cited properly.

We thank the reviewer for pointing out this issue and we have properly cited this reference in the revised manuscript (e.g., in term of the classification of DSK neurons).

Line 104, correct Figure 2I to Figure 1I and line 108, instead of Figure 2J correct to 1J.

We thank the reviewer for pointing out this issue and we have corrected these mistakes.

Line 132 introduces R71G01-GAL4 neurons awkwardly and it seems to come out of nowhere. What was the role of that driver? Why did the authors choose to investigate this GAL4 driver?

We thank the reviewer for pointing out this issue and we have rewritten this part “In males, *R71G01-GAL4* drives the expression of P1 neurons that interact with DSK neurons to regulate male courtship (Wu et al., 2019) and aggression (Wu et al., 2020). Previous studies employed the intersection of *R71G01-LexA* with *dsx^GAL4^* to specifically label and manipulate pC1 neurons, which integrate male courtship and pheromone cues to promote virgin female receptivity (Zhou et al., 2014). We found that activation of *R71G01-GAL4* neurons consisting of pC1 and a few other neurons promoted female receptivity (*Figure 5—figure supplement 1*), similarly as previously activating pC1 neurons using the intersectional strategy (Zhou et al., 2014). Thus, we asked whether DSK neurons would interact with *R71G01-GAL4* neurons to control female sexual behavior.”

Line 175 mentions GluRIA-GAL4 but the reference is missing, please also include the role of this driver.

Indeed, the GAL4 lines including GluRIA-GAL4 used in screening are from Yi Rao’s lab. We have cited the relevant reference in line 172 in the previous manuscript.

As suggested by reviewer, we have rewritten this part as following: “Interestingly, we found that intersection of GluRIA^GAL4^, which targets Glutamate receptor IA (GluRIA) cells, with DskFlp specifically labeled DSK-MP1 neurons (Figure 3A), while intersection of TβH^GAL4^, which targets octopaminergic neurons, with DskFlp specifically labeled DSK-MP3 neurons (Figure 3B).”

Line 191-192, states- 'we constructed knockout lines for these two receptors (figure 6A-C)' but in the figure only one receptor is shown. Are these the same lines as in the earlier study (Wu et al., 2020)?

Yes, the knockout lines used in this study are the same lines as in the earlier study (Wu et al., 2020) and we have changed the description of these lines used in this study.

Line 194-196, RNAi knockdown of CCKLR-17D3 was performed, what is the phenotype from CCKLR-17D1 knock-down?

We have examined the effect on female sexual behavior in females with knocking down the expression of *CCKLR-17D1* and found that knockdown of *CCKLR-17D1* did not affect female receptivity (please see Figure 4—figure supplement 1B).

Line 239, After Dsk- insert 'gene'.

This sentence was removed in the revised manuscript.

Line 267, classification of Dsk neurons, given that Wu et al., Nat. Comm., 2019 study precedes these studies, classification of Dsk neurons and any differences should be mentioned in the main text itself rather than in the Discussion section.

We thank the reviewer for this comment and have described the classification of Dsk neurons in the result section as well as Discussion section in the revised manuscript.

Line 270, for male aggression please also cite Agrawal et al., JEB 2020.

We thank the reviewer’s suggestion and we have added these references in the revised manuscript.

line 488, by RCR do you mean PCR?

Yes, we confirmed the deletion of *Dsk* by PCR analysis at the deletion locus on genomic DNA samples, and the typo has been corrected.

Figure supplements (e.g., Figure 1 figure supplement 6,) repeatedly show the behavioral arena used for courtship assay. This is redundant and should be shown only once and referred to throughout the manuscript as required.

We thank the reviewer’s suggestion and we have made textual changes as suggested.

Reviewer #3 (Recommendations for the authors):I believe the science and its presentation could be strengthened by addressing the points below:Science.1. It is not clear whether the authors have used proper statistical tests. The same flies are monitored over time for copulation rate, which should be considered in the statistical analysis.

We thank the reviewer for this comment and have thoroughly checked all statistical analysis in the revised manuscript.

2. Why have the authors applied a non-parametric test (Χ2) for analyzing latency to copulation (measured in minutes)?

We thank the reviewer for pointing this out and we have re-analyzed these data about latency to copulation and detail statistical test was displayed in the figure legend. (Kruskal-Wallis ANOVA test followed by post-hoc Mann-Whitney U test was used for comparison among multiple groups. The Mann-Whitney U test was applied for analyzing the significance of two columns).

3. Copulation rate and latency to copulation should be presented as different data sets and statistically analyzed independently.

We thank the reviewer for pointing this out and the data sets including copulation rate and latency to copulation have presented independently. In addition, we have been re-analyzed the latency to copulation using proper statistical tests.

4. The y scale of the same type of graphs should be the same across figures.

We have corrected this issue throughout.

5. The scale is truncated in many graphs showing latency to copulation. This should be fixed.

We thank the reviewer for pointing this out and we have corrected this problem in all new figures.

6. The authors should state which experimental groups are compared in each figure. When more than two genotypes are plotted, multiple comparison tests should be applied (e.g., see Figure 2D, latency to copulation; Figure 5D 5F, latency to copulation; Figure 6E, latency to copulation). All comparisons should be reported, either in each graph or in a supp. table.

We thank the reviewer for pointing out this problem. We have been re-analyzed the latency to copulation using proper statistical tests as mentioned above and revised the description in the figure legends.

7. In some figure legends the authors state 'Kruskal-Wallis and post-hoc Mann-Whitney U tests or post-hoc Student's T-test'. For example, in Figure 1-suppl 3. The authors should clearly indicate which test is used to analyze each data set.

We thank the reviewer for pointing this out. We have carefully examined all figure legends about statistical test and clarified the statistical tests used for each data set.

8. What's the rationale for using a post-hoc Student's T-test in Figure 1-suppl 2? A Conover post hoc test might be more appropriate.

We have corrected the statistics in the revised manuscript throughout.

9. The data strongly support a role for DSK and DSK neurons control female sexual receptivity. However, it is unclear whether DSK neurons in the VNC contribute to the observed phenotypes. An experiment addressing this point would further support the findings of the study.

Indeed, by using *Dsk^GAL4^* to drive the expression of *UAS-GFP,* we found that there no DSK neurons in the VNC and only have projections from brain by using *Dsk^GAL4^* to drive the expression of *UAS-GFP* (please see Figure 1—figure supplement 3D). Thus, we think that DSK neurons in the brain play a role in controlling female receptivity.

10. Figure 1 A—figure supplement 4. None of the control virgin females have laid any unfertilized eggs across 3 days, which is unexpected.

We indeed observed that the virgin females did not lay any unfertilized eggs within 3 days and it may be because that the females used to test the egg laying behavior were too young. To ensure that the age of females is the same between egg laying experiment and mating behavior, we re-tested egg laying behavior within 3 days after the indicated genotype of virgin females aging 4 days and found that mutating of *Dsk* or knocking down the expression of *Dsk* did not affect egg laying behavior of virgin females (please see Figure 1—figure supplement 6).

11. Figure 1B-suppl 4. More details should be added in the graph and legend to understand the TRIC data.

We thank the reviewer for this suggestion. We have added more details in the graph and legend.

12. Do the authors have data showing which aspects of female sexual behavior are regulated by the DSK signaling? For example, does activation of DSK neurons induce any female acceptance behaviors (e.g., partial ovipositor extrusion)? Perhaps DSK neurons trigger motor outputs associated with sexual receptivity? If DSK neurons acted downstream of pC1 neurons, we would expect them to induce vaginal plate opening. These possibilities should be experimentally addressed or discussed in the manuscript.

We thank the reviewer for this suggestion. We re-recorded high resolution videos. We asked whether the phenotype of decreased female receptivity in *Dsk* mutant flies is due to potentially elevated ovipositor extrusion (a rejection behavior). However, we found that manipulating *Dsk* gene did not affect ovipositor extrusion (please see Figure 1—figure supplement 2). Similarly, we also analyzed whether activating DSK neurons would affect ovipositor extrusion in females with courting males. We found that manipulation of DSK neurons did not affect ovipositor extrusion (please see Figure 2—figure supplement 1). Finally, we further discussed the potential effect of Dsk signaling on female sexual behavior (*e.g.*, potential role of pausing behavior to male courtship) in the Discussion section.

13. I believe more information about the expression pattern of Dsk in neuronal and non-neuronal tissues would help better understand the findings. Details of the number of DSK neurons, different brain clusters, as well as projection patterns, would help the authors to put their findings in context. The authors should add confocal images showing the expression pattern of Dsk neurons in VNC.A better description of the expression pattern of Dsk would improve the narrative and help the authors better explain (1) the anatomical and functional experiments linking DSK neurons with 71G01-Gal4 neurons (2) the identification of subsets of DSK clusters key for sexual receptivity (3) the importance of manipulating DSK and Dsk-expressing cells solely in the nervous system.

We thank the reviewer for this helpful suggestion. Following the reviewer’s suggestion, we have provided detail descriptions of the DSK neurons in the revised manuscript. In addition, we also have added the expression pattern of DSK neurons in the VNC in the revised manuscript (please see Figure 1—figure supplement 3D). Indeed, this Dsk^GAL4^ line did not label neurons in the ventral nerve cord or the gut.

14. Dsx-expressing neurons have a prominent role in controlling female sexual receptivity. Is there any indication that DSK neurons and Dsx neurons fully/partially overlap?

We have examined whether DSK neurons are *dsx*^+^ neurons by using intersectional technology and found that DSK neurons (magenta) did not overlap with *dsx* neurons (green).

15. I believe Table 1 is missing in the manuscript.

We have added Table 1 in the revised manuscript. We are sorry for this mistake.

16. Some genetic controls are missing in several figures. For instance, in Figure 2, elav-GAL4/+ is missing.

We thank the reviewer for this suggestion and have added the control line (elav-GAL4/+) in Figure 1J (please see Figure 1J).

17. The authors should clarify that the GRASP experiments don't provide information about the direction of connectivity.

We thank the reviewer for pointing out this and we have rewritten the sentence to only suggest such possibility.

18. The authors should add images showing the pattern of DSK neurons targeted with genetic intersections using GluRIA-Gal4 and TbH-Gal4 with DSK-FLP in the VNC.

We thank the reviewer for this suggestion and have mapped DSK neurons in the VNC as suggested. We found that there was no cell body in the VNC and only some projection from the brain (please see Figure 3—figure supplement 1).

Presentation:I believe a substantial improvement in the writing style would help readers fully judge and appreciate the findings of this interesting study.1. In my opinion, the opening paragraph of Introduction is not engaging. Overall, I think the authors could strengthen the Introduction by adding information about the courtship ritual in flies and explaining its relevance for mate selection. Moreover, they could describe how female flies signal sexual receptivity and accept a male for copulation. This would help them highlight the importance of their work and make it more attractive for non-specialist readers.

We thank the reviewer for these valuable suggestions. We have now expanded our introduction as suggested.

2. My impression is that the Results section reads like a long list of results, which are not connected to tell an interesting story. The rationale of many of the experiments are either presented in the Discussion section or not discussed at all. The authors could speculate how DSK neurons/ DSK might work to regulate sexual receptivity to introduce subsequent experiments in the main text. A better description of the results and their potential implications would help the narrative of the study (more details below).

Thanks for the suggestions. In the revised manuscript, we have explained the rationale of each experiment and more accurately described the results.

3. One of the strengths of this study is the arsenal of modern tools used to support the findings. However, many of the methods are not described in the main text. More information would help readers to fully grasp the approaches and findings of this study.

We thank the reviewer for these suggestions. We have made textual change as suggestion.

4. The Discussion section superficially discusses the findings of the study. It rather reads like a recapitulation of the results. The authors could enrich the discussion by speculating how DSK neurons might control female sexual receptivity. Could pC1 neurons (targeted by 71G01-Gal4?) act as a central node for female sexual receptivity, conveying female mating decisions to Dsk neurons? Alternatively, could DSK neurons also integrate sensory information relevant to sexual behavior? Are there any sensory pathways relevant to female sexual receptivity feeding into DSK neurons? What are the potential downstream targets of DSK? What aspects of female sexual behavior are regulated by the DSK pathway? Do females show increased vaginal plate/ partial ovipositor extrusion? How do DSK neurons differ between males and females, and how do these differences may result in different sex-specific behaviors?

We thank the reviewer for these valuable suggestions. We have now expanded our discussion part as suggested.

5. The schematic in Figure 4 A is not clear.

We are sorry that the schematics in Figure 4A which did not clearly present our aim in our electrophysiological experiments, and we have re-organized these schematics and made explained in greater detail in the figure legend (please see Figure 6A).

6. Figure 4. A bigger schematic of the brain depicting the different DSK clusters would help understand which neurons are assessed.

We thank the reviewer for this suggestion and have enlarged the brain.

7. The authors should make the figures accessible to readers with color-blindness.

We thank the reviewer for pointing out this issue and we have adjusted the color in the revised manuscript.

8. I suggest the authors proofread the manuscript, as there are typos, grammatical mistakes and unclear sentences throughout the manuscript. Some examples:Line 46 – 'is' should be changed to 'are'.

Corrected.

Line 87 – This sentence contains grammatical mistakes.

We have rewritten the sentence “Two parameters including copulation rate and latency were used to characterize receptivity”.

Line 93 – RNAi-mediated females: this should be better defined.

We have changed “RNAi-mediated females” to “we knocked down the expression of *Dsk* using RNA interference (RNAi)…”.

Line 109 – 'Indicated' should be changed to 'indicate'.

Corrected.

Line 104 – Figure is wrongly cited. It should be Figure 1I.

Corrected.

Line 112 – 'were' should be changed to 'are'.

Corrected.

Line 132 – 'expressed' should be changed to 'are present'.

Corrected.

Line 134 -Fix this sentence 'Given that, we hypothesized whether DSK'.

We have rewritten the sentence “Thus, we asked whether DSK neurons would interact with R71G01-GAL4 neurons to control female sexual behavior.”

Line 139 – Change 'recombinant' to 'reconstituted'.

Corrected.

Line 157 -The sentence does not make sense.

Removed.

Line 169 – Unclear sentence.

We have rewritten the sentence “To investigate whether one or both of the types are involved in regulating female sexual behavior…”.

Line 188 – It should be-'wanted' not 'want'.

Corrected.

Line 270 – The sentence should be rewritten: In this study, DSK neurons also modulate female sexual behavior.

We have removed this sentence.

Line 488 – Fix this sentence 'RCR analysis from genomic DNA samples'.

We have rewritten the sentence “…PCR analysis at the deletion locus on genomic DNA samples of *ΔDsk/ΔDsk*, *+/ΔDsk*, *+/+*”.

Line 502 – Fix this sentence: 'And there are same numbers in two parameters'.

This sentence has been removed and the number of female flies paired with wide-type males is shown in figures.

'Latency' should be described as' latency to copulation'.

We have replaced ‘Latency’ with ‘latency to copulation’ in the revised manuscript throughout.

Figure 1.Suppl 5 – fix the typo in the title.The authors should better articulate the rationale of the experiments and explain the techniques so readers can better appreciate the findings. For example:Line 86 – 'behavior, we first constructed knock out line for Dsk'. Describe how the knock out line for DsK was made.

We thank the reviewer for suggestion and we have changed the description of this line used in this study and also briefly explained their generation as following: “*.. Dsk* mutant *(∆Dsk),* which was generated previously (Wu et al., 2020). In brief, the 5’-UTR and coding region were deleted by the CRISPR-Cas9 system”.

Line 94 – Explain the rationale for looking at male courtship behavior.

We have rewritten this part as following: “To investigate whether reduced copulation rate in ∆*Dsk* females is due to potential abatement of female sexual appeal, we examined courtship levels in wild-type males paired with ∆*Dsk* or control females and observed similarly high levels of courtship in all cases …”.

Line 96 – Explain the rationale for looking at egg laying.

We have rewritten this part as following: “As recently mated females may reduce receptivity and increase egg-laying, we asked whether the decreased receptivity could be a post-mating response and correlate with elevated egg-laying. To address this, we examined the number of eggs laid by virgin females with *Dsk* mutant or knockdown….”.

Line 97 – Explain the rationale of looking at TRIC signal changes. The authors should also explain how this technique works and provide more details on the findings.

We have rewritten this part as following: “To investigate whether DSK neurons respond to mating status, we measured the activity of these neurons using the transcriptional reporter of intracellular Ca^2+^ (TRIC) in virgin and mated females. TRIC is designed to quantitatively monitor the change of neural activity by the reconstituted of a functional transcription factor in the presence of Ca^2+^ (Gao et al., 2015). As above mentioned, four pairs of neurons were labeled by *Dsk^GAL4^* driving the expression of *UAS-mCD8::GFP* (*Figure 1—figure supplement 3B*). However, we only observed TRIC signals in four neurons in the middle area of female brains (*Figure 1—figure supplement 6B,C*). Quantification of these TRIC signals showed no significant difference in virgin and mated females (*Figure 1—figure supplement 6D*). These results further indicate that DSK neurons do not respond to mating status.”. we also provided more details in the corresponding figures and figure legends.

The authors should clearly define if any of the tools used in this study were previously generated.

We thank the reviewer for this suggestion. Indeed, a few reagents were generated in our previous *eLife* paper (Wu et al., 2020). In the revised manuscript, we clearly indicated whether the genetic reagent was generated in this study or a previous study.

Line 102 – Define the DsK Gal4 used in Figure 1I. Is it the same line used in subsequent Figures?

Yes, the Dsk-GAL4 line used in Figure 1I is the same as the line used in the subsequent Figures. And we changed the text in the revised manuscript.

Line 109 – The conclusion should also state that the evidence points to a role of DSK neurons in female sexual receptivity.

We thank the reviewer for this suggestion and have rewritten the sentence “Taken together, these results indicated the function of *Dsk* is crucial for female sexual receptivity, which also suggested DSK neurons play a role in female sexual receptivity”.

Line 119 – Explain what's the rationale for looking at very young females. State their age and differences with previous virgin females tested in other experiments.

We have rewritten this part as following: “Female receptivity depends on the female’s sexual maturity and mating status. Very young virgins display low receptivity level to courting males and mated females become temporarily unreceptive to courting males (Dickson, 2008; Rezaval et al., 2012). We tested whether activation of DSK neurons could also promote female sexual receptivity in very young virgins or mated females…”.

Line 133 – More information about 71G01-Gal4, and the fact that it might intersect pC1 neurons, would help explain the rationale behind testing these neurons.

We thank the reviewer for this suggestion and have re-stated this part as suggested. “In males, *R71G01-GAL4* drives the expression of P1 neurons that interact with DSK neurons to regulate male courtship (Wu et al., 2019) and aggression (Wu et al., 2020). Previous studies employed the intersection of *R71G01-LexA* with *dsx^GAL4^* to specifically label and manipulate pC1 neurons, which integrate male courtship and pheromone cues to promote virgin female receptivity (Zhou et al., 2014). We found that activation of *R71G01-GAL4* neurons consisting of pC1 and a few other neurons promoted female receptivity (*Figure 5—figure supplement 1*), similarly as previously activating pC1 neurons using the intersectional strategy (Zhou et al., 2014)…”.

Line 157 – Here it is important to explain how many DSK neurons are present in the brain, and how they are distributed in different clusters (by including images of the nervous system).

We thank the reviewer for this suggestion and we have added this information as suggested in the revised manuscript as following: “Analyses of the expression pattern of *Dsk^GAL4^* revealed that four pairs of neurons were specifically labeled in the brain, which were classified into two types (two pairs of MP1 and two pairs of MP3) based on the location of cell bodies (Nichols and Lim, 1996) (*Figure 1—figure supplement 3D*), and the two pairs of MP1 neurons were further classified into MP1a and MP1b based on the single cell morphology of these neurons (Wu et al., 2019)”.

Line 175 – The authors identify two Gal4 lines (GluRIAGAL4 and TβHGAL4) that help them intersect different DSK neurons but these lines are not described. The authors should explain what neurons are targeted by these drivers (e.g., glutamatergic, octopaminergic, etc). This information could be useful to interpret and discuss their findings (e.g., relevant DSK neurons might be excitatory).

We thank the reviewer for this suggestion and have made textual changes as following: “Interestingly, we found that intersection of GluRIA^GAL4^, which targets Glutamate receptor IA (GluRIA) cells, with DskFlp specifically labeled DSK-MP1 neurons (Figure 3A), while intersection of TβH^GAL4^, which targets octopaminergic neurons, with DskFlp specifically labeled DSK-MP3 neurons (Figure 3B)”.

Line 164 – Explain how the P2X2 optogenetic approach works.

We think that this is a misunderstanding for our electrophysiological recording experiments and may be due to unclear schematics. Indeed, we activated R71G01-GAL4 neurons through ATP activation of ATP-gated P2X_2_ channel and recorded the electrical responses in DSK-MP1 neurons and DSK-MP3 neurons. In addition, we have re-organized our schematics and added explanation in the figure legend in the revised manuscript.

Line 191 – 'We constructed knock out lines for these two receptors (Figure 6A-C) (Wu et al., 2020)'. Clearly state if these are novel tools created for this study.

We thank the reviewer for suggestion and have changed the description of these lines used in this study and explained their generation as following: “…*CCKLR-17D3* mutant female, which was also generated previously (Wu et al., 2020). In brief, the last four exons were deleted by the CRISPR-Cas9 system …*.*”.